# Spatial-Temporal Super-Resolution of Satellite Imagery via Conditional Pixel Synthesis

**Yutong He**     **Dingjie Wang**     **Nicholas Lai**     **William Zhang**     **Chenlin Meng**
**Marshall Burke**     **David B. Lobell**     **Stefano Ermon**
Stanford University
{kellyyhe, daviddw, nicklai, wxyz, chenlin, ermon}@cs.stanford.edu
{mburke, dlobell}@stanford.edu

## Abstract

High-resolution satellite imagery has proven useful for a broad range of tasks, including measurement of global human population, local economic livelihoods, and biodiversity, among many others. Unfortunately, high-resolution imagery is both infrequently collected and expensive to purchase, making it hard to efficiently and effectively scale these downstream tasks over both time and space. We propose a new conditional pixel synthesis model that uses abundant, low-cost, low-resolution imagery to generate accurate high-resolution imagery at locations and times in which it is unavailable. We show that our model attains photo-realistic sample quality and outperforms competing baselines on a key downstream task – object counting – particularly in geographic locations where conditions on the ground are changing rapidly.

## 1   Introduction

Recent advancements in satellite technology have enabled granular insight into the evolution of human activity on the planet's surface. Multiple satellite sensors now collect imagery with spatial resolution less than 1m, and this high-resolution (HR) imagery can provide sufficient information for various fine-grained tasks such as post-disaster building damage estimation, poverty prediction, and crop phenotyping [15, 3, 41]. Unfortunately, HR imagery is captured infrequently over much of the planet's surface (once a year or less), especially in developing countries where it is arguably most needed, and was historically captured even more rarely (once or twice a decade) [7]. Even when available, HR imagery is prohibitively expensive to purchase in large quantities. These limitations often result in an inability to scale promising HR algorithms and apply them to questions of broad social importance. Meanwhile, multiple sources of publicly-available satellite imagery now provide sub-weekly coverage at global scale, albeit at lower spatial resolution (e.g. 10m resolution for Sentinel-2). Unfortunately, such coarse spatial resolution renders small objects like residential buildings, swimming pools, and cars unrecognizable.

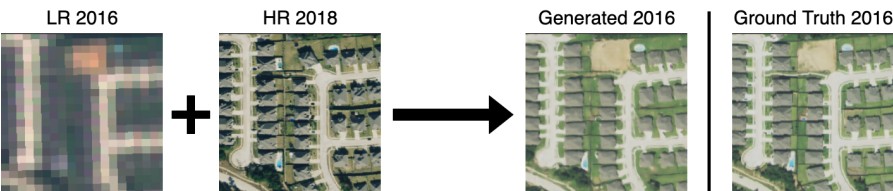

Figure 1: Given a 10m low resolution (LR) image from 2016 and a 1m high resolution (HR) image from 2018, we generate a photo-realistic and accurate HR image for 2016.

35th Conference on Neural Information Processing Systems (NeurIPS 2021).

In the last few years, thanks to advances in deep learning and generative models, we have seen great progress in image processing tasks such as image colorization [43], denoising [6, 35], inpainting [35, 27], and super-resolution [11, 21, 16]. Furthermore, pixel synthesis models such as neural radiance field (NeRF) [26] have demonstrated great potential for generating realistic and accurate scenes from different viewpoints. Motivated by these successes and the need for high-resolution images, we ask whether it is possible to synthesize high-resolution satellite images using deep generative models. For a given time and location, can we generate a high-resolution image by interpolating the available low-resolution and high-resolution images collected over time?

To address this question, we propose a conditional pixel synthesis model that leverages the fine-grained spatial information in HR images and the abundant temporal availability of LR images to create the desired synthetic HR images of the target location and time. Inspired by the recent development of pixel synthesis models pioneered by the NeRF model [26, 40, 2], each pixel in the output images is generated conditionally independently by a perceptron-based generator given the encoded input image features associated with the pixel, the positional embedding of its spatial-temporal coordinates, and a random vector. Instead of learning to adapt to different viewing directions in a single 3D scene [26], our model learns to interpolate across the time dimension for different geo-locations with the two multi-resolution satellite image time series.

To demonstrate the effectiveness of our model, we collect a large-scale paired satellite image dataset of residential neighborhoods in Texas using high-resolution NAIP (National Agriculture Imagery Program, 1m GSD) and low-resolution Sentinel-2 (10m GSD) imagery. This dataset consists of scenes in which housing construction occurred between 2014 and 2017 in major metropolitan areas of Texas, with construction verified using CoreLogic tax and deed data. These scenes thus provide a rapidly changing environment on which to assess model performance. As a separate test, we also pair HR images (0.3m to 1m GSD) from the Functional Map of the World (fMoW) dataset [9] crop field category with images from Sentinel-2.

To evaluate our model's performance, we compare to state-of-the-art methods, including super-resolution models. Our model outperforms all competing models in sample quality on both datasets measured by both standard image quality assessment metrics and human perception (see example in Figure 1). Our model also achieves $0.92$ and $0.62$ Pearson's $r^2$ in reconstructing the correct numbers of buildings and swimming pools respectively in the images, outperforming other models in these tasks. Results suggest our model's potential to scale to downstream tasks that use these object counts as input, including societally-important tasks such as population measurement, poverty prediction, and humanitarian assessment [7, 3].

## 2   Related Work

**Image Super-resolution**   SRCNN [11] is the first paper to introduce convolutional layers into a SR context and demonstrate significant improvement over traditional SR models. SRGAN [21] improves on SRCNN with adversarial loss and is widely compared among many GAN-based SR models for remote sensing imagery [36, 25, 29]. DBPN [16] is a state-of-the-art SR solution that uses an iterative algorithm to provide an error feedback system, and it is one of the most effective SR models for satellite imagery [28]. However, [31] shows that SR is less beneficial at coarser resolution, especially when applied to downstream object detection on satellite imagery. In addition, most SR models test on benchmarks where LR images are artificially created, instead of collected from actual LR devices [1, 39, 17]. SR models also generally perform worse at larger scale factors, which is closer to settings for satellite imagery SR in real life.

SRNTT [45] applies reference-based super-resolution through neural texture transfer to mitigate information loss in LR images by leveraging texture details from HR reference images. While SRNTT also uses a HR reference image, it does not learn the additional time dimension to leverage the HR image of the same object at a different time. In addition, our model uses a perceptron based generator while SRNTT uses a CNN based generator.

**Fusion Models for Satellite Imagery**   [12] first proposes STARFM to blend data from two remote sensing devices, MODIS [4] and Landsat [20], for spatial-temporal super resolution to predict land reflectance. [46] introduces an enhanced algorithm for the same task and [10] combines linear pixel unmixing and STARFM to improve spatial details in the generated images. cGAN Fusion

[5] incorporates GAN-based models in the solution, using an architecture similar to Pix2Pix [18]. In contrast to previous work, we are particularly interested in synthesizing images with very high resolution ($\leq$ 1m GSD), enabling downstream applications such as poverty level estimation.

**NeRF and Pixel Synthesis Models**  Recent developments in deep generative models, especially advances in perceptron-based generators, have yet to be explored in remote sensing applications. Introduced by [26], neural radiance fields (NeRF) demonstrates great success in constructing 3D static scenes. [23, 38] extends the notion of NeRF and incorporates time-variant representations of the 3D scenes. [30] embeds NeRF generation into a 3D aware image generator. These works, however, are limited to generating individual scenes, in contrast with our model which can generalize to different locations in the dataset. [40] proposes a framework that predicts NeRF conditioning on spatial features from input images; however, it requires constructing the 3D scenes, which is less applicable to satellite imagery. [2] proposes a style-based 2D image generative model using an only perceptron-based architecture; however, unlike our method, it doesn't consider the task of conditional 2D image generation nor does it incorporate other variables such as time. In contrast, we propose a pixel synthesis model that learns a conditional 2D spatial coordinate grid along with a continuous time dimension, which is tailored for remote sensing, where the same location can be captured by different devices (e.g. NAIP or Sentinel-2) at different times (e.g. year 2016 or year 2018).

## 3   Problem Setup

The goal of this work is to develop a method to synthesize high-resolution satellite images for locations and times for which these images are not available. As input we are given two time-series of high-resolution (HR) and low-resolution (LR) images for the same location. Intuitively, we wish to leverage the rich information in HR images and the high temporal frequency of LR images to achieve the best of both worlds.

Formally, let $I_{hr}^{(t)} \in \mathbf{R}^{C \times H \times W}$ be a sequence of random variables representing high-resolution views of a location at various time steps $t \in T$. Similarly, let $I_{lr}^{(t)} \in \mathbf{R}^{C \times H_{lr} \times W_{lr}}$ denote low-resolution views of the same location over time. Our goal is to develop a method to estimate $I_{hr}^{(t)}$, given $K$ high resolution observations $\{I_{hr}^{(t'_1)}, \cdots, I_{hr}^{(t'_K)}\}$ and $L$ low-resolution ones $\{I_{lr}^{(t''_1)}, \cdots, I_{lr}^{(t''_L)}\}$ for the same location. Note the available observations could be taken either before or after the target time $t$. Our task can be viewed as a special case of multivariate time-series imputation, where two concurrent but incomplete series of satellite images of the same location in different resolutions are given, and the model should predict the most likely pixel values at an unseen time step of one image series.

In this paper, we consider a special case where the goal is to estimate $I_{hr}^{(t)}$ given a single high-resolution image $I_{hr}^{(t')}$ and a single low-resolution image $I_{lr}^{(t)}$ also from time $t$. We focus on this special case because while typically $L \gg K$, it is reasonable to assume $I_{hr}^{(t)} \perp\!\!\!\perp I_{lr}^{(t')} \mid I_{lr}^{(t)}$ for $t' \neq t$, i.e., given a LR image at the target time $t$, other LR views from different time steps provide little or no additional information. Given the abundant availability of LR imagery, it is often safe to assume access to $I_{lr}^{(t)}$ at target time $t$. Figure 1 provides a visualization of this task.

For training, we assume access to paired triplets $\{I_{hr}^{(t)}, I_{lr}^{(t)}, I_{hr}^{(t')}\}$ collected across a geographic region of interest where $t' \neq t$. At inference time, we assume availability for $I_{lr}^{(t)}$ and $I_{hr}^{(t')}$ and the model needs to generalize to previously unseen locations. Note that at inference time, the target time $t$ and reference time $t'$ may not have been seen in the training set either.

## 4   Method

Given $I_{lr}^{(t)}$ and $I_{hr}^{(t')}$ of the target location and target time $t$, our method generates $\hat{I}_{hr}^{(t)} \in \mathbf{R}^{C \times H \times W}$ with a four-module conditional pixel synthesis model. Figure 2 is an illustration of our framework.

The generator $G$ of our model consists of three parts: image feature mapper $F : \mathbf{R}^{C \times H \times W} \to \mathbf{R}^{C_{fea} \times H \times W}$, positional encoder $E$, and the pixel synthesizer $G_p$. For each spatial coordinate $(x, y)$ of the target HR image, the image feature mapper extracts the neighborhood information around

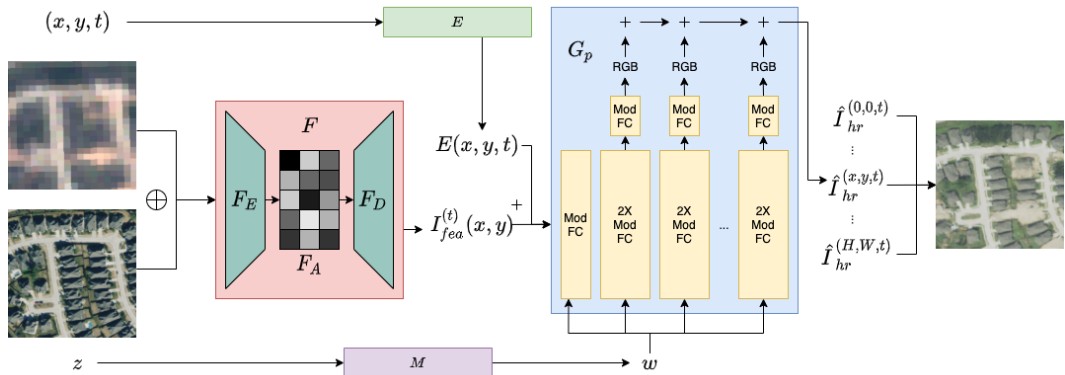

Figure 2: An illustration of our proposed framework (discriminator omitted). The input images are processed by the image feature mapper $F$ to obtain $I_{fea}^{(t)}$. Then with its spatial-temporal coordinate $(x, y, t)$ encoded by $E$, each pixel is synthesized conditionally independently given the image feature associated with its spatial coordinate $I_{fea}^{(t)}(x, y)$ and a random vector $z$.

$(x, y) \in \{0, 1, ..., H\} \times \{0, 1, ..., W\}$ from $I_{lr}^{(t)}$ and $I_{hr}^{(t')}$, as well as the global information associated with the coordinate in the two input images. The positional encoder learns a representation of the spatial-temporal coordinate $(x, y, t)$, where $t$ is the temporal coordinate of the target image. The pixel synthesizer then uses the information obtained from the image feature mapper and the positional encoding to predict the pixel value at each coordinate. Finally, we incorporate an adversarial loss in our training, and thus include a discriminator $D$ as the final component of our model.

**Image Feature Mapper**   Before extracting features, we first perform nearest neighbor resampling to the LR image $I_{lr}^{(t)}$ to match the dimensionality of the HR image and concatenate $I_{lr}^{(t)}$ and $I_{hr}^{(t')}$ along the spectral bands to form the input $I_{cat}^{(t)} = \text{concat}[I_{lr}^{(t)}, I_{hr}^{(t')}] \in \mathbf{R}^{2C \times H \times W}$. Then the mapper processes $I_{cat}^{(t)}$ with a neighborhood encoder $F_E : \mathbf{R}^{2C \times H \times W} \to \mathbf{R}^{C_{fea} \times H' \times W'}$, a global encoder $F_A : \mathbf{R}^{C_{fea} \times H' \times W'} \to \mathbf{R}^{C_{fea} \times H' \times W'}$ and a neighborhood decoder $F_D : \mathbf{R}^{C_{fea} \times H' \times W'} \to \mathbf{R}^{C_{fea} \times H \times W}$. The neighborhood encoder and decoder learn the fine structural features of the images, and the global encoder learns the overall inter-pixel relationships as it observes the entire image.

$F_E$ uses sliding window filters to map a small neighborhood of each coordinate into a value stored in the neighborhood feature map $I_{ne}^{(t)} \in \mathbf{R}^{C_{fea} \times H' \times W'}$ and $F_D$ uses another set of filters to transform the global feature map $I_{gl}^{(t)} \in \mathbf{R}^{C_{fea} \times H' \times W'}$ back to the original coordinate grid. $F_A$ is a self-attention module that takes $I_{ne}^{(t)}$ as the input and learns functions $Q, K : \mathbf{R}^{C_{fea} \times H' \times W'} \to \mathbf{R}^{C_{fea}/8 \times H'W'}, V : \mathbf{R}^{C_{fea} \times H' \times W'} \to \mathbf{R}^{C_{fea} \times H'W'}$ and a scalar parameter $\gamma$ to map $I_{ne}^{(t)}$ to $I_{gl}^{(t)}$.

The image feature mapper $F = F_E \circ F_A \circ F_D$ and we denote $I_{fea}^{(t)} = F(I_{cat}^{(t)})$ and the image feature associated with coordinate $(x, y)$ as $I_{fea}^{(t)}(x, y) \in \mathbf{R}^{C_{fea}}$. Details are available in Appendix A.

**Positional Encoder**   Following [2], we also include both the Fourier feature and the spatial coordinate embedding in the positional encoder $E$. The Fourier feature is calculated as $e_{f_o}(x, y, t) = \sin(B_{f_o}(\frac{2x}{H-1} - 1, \frac{2y}{H-1} - 1, \frac{t}{u}))$ where $B_{f_o} \in \mathbf{R}^{3 \times C_{fea}}$ is a learnable matrix and $u$ is the time unit. This encoding of $t$ allows our model to handle time-series with various lengths and to extrapolate to time steps that are not seen at training time. $E$ also learns a $C_{fea} \times H \times W$ matrix $e_{co}$ and the spatial coordinate embedding for $(x, y, t)$ is extracted from the vector at $(x, y)$ in $e_{co}$. The positional encoding of $(x, y, t)$ is the channel concatenation of $e_{f_o}(x, y, t)$ and $e_{co}(x, y, t)$, $E(x, y, t) = \text{concat}[e_{f_o}(x, y, t), e_{co}(x, y, t)] \in \mathbf{R}^{2C_{fea}}$.

**Pixel Synthesizer**   Pixel Synthesizer $G_p$ can be viewed as an analogy of simulating a conditional $2 + 1D$ neural radiance field with fixed viewing direction and camera ray using a perceptron based

model. Instead of learning the breadth representation of the location, $G_p$ learns to scale in the time dimension in a fixed spatial coordinate grid. Each pixel is synthesized conditionally independently given $I_{fea}^{(t)}$, $E(x, y, t)$, and a random vector $z \in \mathbf{R}^Z$. $G_p$ first learns a function $g_z$ to map $E(x, y, t)$ to $\mathbf{R}^{C_{fea}}$, then obtains the input to the fully-connected layers $e(x, y, t) = g_z(E(x, y, t)) + I_{fea}^{(t)}(x, y)$. Following [2, 19], we use a $m$-layer perceptron based mapping network $M$ to map the noise vector $z$ into a style vector $w$, and use $n$ modulated fully-connected layers (ModFC) to inject the style vector into the generation to maintain style consistency among different pixels of the same image. We map the intermediate features to the output space for every two layers and accumulate the output values as the final pixel output.

With all components combined, the generated pixel value at $(x, y, t)$ can be calculated as

$$\hat{I}_{hr}^{(t)}(x, y) = G(x, y, t, z | I_{lr}^{(t)}, I_{hr}^{(t')}) = G_p(E(x, y, t), F(I_{cat}^{(t)}), z)$$

**Loss Function**    The generator is trained with the combination of the conditional GAN loss and $L_1$ loss. The objective function is

$$G^* = \arg \min_G \max_D \mathcal{L}_{cGAN}(G, D) + \lambda \mathcal{L}_{L_1}(G)$$

$\mathcal{L}_{cGAN}(G, D) = \mathbb{E}[\log D(I_{hr}^{(t)}, X, I_{lr}^{(t)}, I_{hr}^{(t')})] + \mathbb{E}[1 - \log D(G(X, z | I_{lr}^{(t)}, I_{hr}^{(t')}), X, I_{lr}^{(t)}, I_{hr}^{(t')})]$ where $X$ is the temporal-spatial coordinate grid $\{(x, y, t) | 0 \le x \le H, 0 \le y \le W\}$ for $I_{hr}^{(t)}$. $\mathcal{L}_{L_1}(G) = \mathbb{E}[||I_{hr}^{(t)} - G(X, z | I_{lr}^{(t)}, I_{hr}^{(t')})||_1]$.

## 5 Experiments

### 5.1 Datasets

**Texas Housing Dataset**    We collect a dataset consisting of 286717 houses and their surrounding neighborhoods from CoreLogic tax and deed database that have an effective year built between 2014 and 2017 in Texas, US. We reserve 14101 houses from 20 randomly selected zip codes as the testing set and use the remaining 272616 houses from the other 759 zip codes as the training set. For each house in the dataset, we obtain two LR-HR image pairs, one from 2016 and another from 2018. In total, there are 1146868 multi-resolution images collected from different sensors for our experiments. We source high resolution images from NAIP (1m GSD) and low resolution images from Sentinel-2 (10m GSD) and only extract RGB bands from Google Earth Engine [14]. More details can be found in Appendix C.

**FMoW-Sentinel2 Crop Field Dataset**    We derive this dataset from the crop field category in Functional Map of the World (fMoW) dataset [9] for the task of generating images over a greater number of time steps. We pair each fMoW image with a lower resolution Sentinel-2 RGB image captured at the same location and a similar time. We prune locations with fewer than 2 timestamps, yielding 1752 locations and a total of 4898 fMoW-Sentinel2 pairs. Each location contains between 2-15 timestamps spanning from 2015 to 2017. We reserve 237 locations as the testing set and the remaining 1515 locations as the training set. More details can be found in Appendix C.

### 5.2 Implementation Details

**Model Details**    We choose $H = W = 256$, $C = 3$ (the concatenated RGB bands of the input images), $C_{fea} = 256$, $m = 3$, $n = 14$ and $\lambda = 100$ for all of our experiments. We use non-saturating conditional GAN loss for $G$ and $R_1$ penalty for $D$, which has the same network structure as the discriminator in [19, 2]. We train all models using Adam optimizer with learning rate $2 \times 10^{-3}, \beta_0 = 0, \beta_1 = 0.99, \epsilon = 10^{-8}$. We train each model to convergence, which takes around 4-5 days on 1 NVIDIA Titan XP GPU. Further details can be found in Appendix A and B.

We provide two versions of the image feature mapper. In version "EAD", we use convolutional layers and transpose convolutional layers with stride $> 1$ in $F_E$ and $F_D$. In version "EA", we use convolutional layers with stride $= 1$ in $F_E$ and an identity function in $F_D$. For version "EA", we use a patch-based training and inference method with a patch size of $64$ because of memory constraints,

and denote it as "EA64". The motivation for including both "EAD" and "EA" is to examine the capabilities of $F$ with and without spatial downsampling or upsampling. "EAD" can sample 1500 images in around 2.5 minutes (10 images/s) and "EA64" can sample 1500 images in around 19 minutes (1.3 image/s). More details can be found in Appendix A and B.

**Baselines** We compare our method with two groups of baseline methods: image fusion models and super-resolution (SR) models. We use cGAN Fusion [5], which leverages the network structure of the leading image-to-image translation model Pix2Pix [18] to combine different imagery products for surface reflectance prediction. We also compare our model with the original Pix2Pix framework. For SR baselines, we choose SRGAN [21], which is widely compared among other GAN based SR models for satellite imagery [36, 25, 29]. We also compare our method with DBPN [16], which is a state-of-the-art SR model for satellite imagery [28].

## 5.3 Image Generation Quality

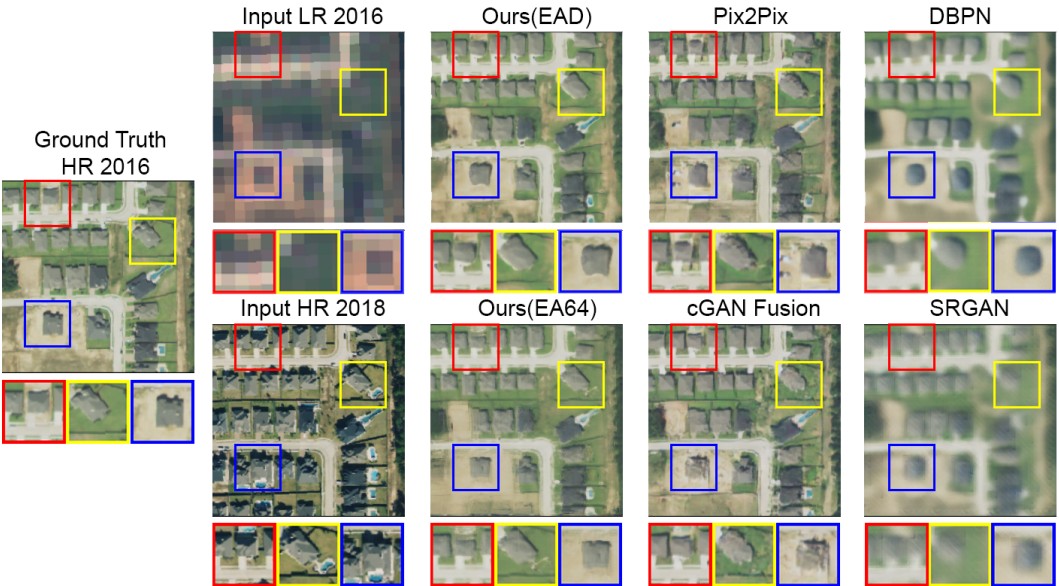

Figure 3: Samples from all models on the Texas housing dataset with setting $t' > t$. Our models show advantages in both sample quality and structural detail consistency with the ground truth, especially in areas with house or pool construction (zoomed in with colored boxes).

We examine generated image quality using both our Texas housing dataset and our fMoW-Sentinel2 crop field dataset. Figures 3 and 4 present qualitative results from our approach and from baselines. Table 1 shows quantitative results on the Texas housing dataset and Table 2 shows quantitative results on the fMoW-Sentinel2 crop field dataset. Overall, our models outperform all baseline approaches in all evaluation metrics.

**Evaluation Metrics** To assess image generation quality, we report standard sample quality metrics SSIM [37], FSIM [42], and PSNR to quantify the visual similarity and pixel value accuracy of the generated images. We also include LPIPS [44] using VGG [32] features, which is a deep perceptual metric used in previous works to evaluate generation quality in satellite imagery [13]. LPIPS leverages visual features learned in deep convolutional neural networks that better reflect human perception. We report the average score of each metric given every possible pair of $t, t'$ where $t \neq t'$.

**Texas Housing Dataset** The dataset focuses on regions with residential building construction between 2014 and 2017, so we separately analyze the task to predict $\hat{I}_{hr}^{(t)}$ when $t' > t$ and when $t' < t$ on the Texas housing dataset. $t' > t$ represents the task to "rewind" the construction of the neighborhood and $t' < t$ represents the task to "predict" construction. Our models achieve more photo-realistic sample quality (measured by LPIPS), maintain better structural similarity (measured

Table 1: Image sample quality quantitative results on Texas housing data. $t' > t$ denotes the task for generating an image in the past given a future HR image, and $t' < t$ denotes the task for generating an image in the future given a past HR image.

| Model | $t' > t$ | | | | $t' < t$ | | | |
|---|---|---|---|---|---|---|---|---|
| | SSIM↑ | PSNR↑ | FSIM↑ | LPIPS↓ | SSIM↑ | PSNR↑ | FSIM↑ | LPIPS↓ |
| Pix2Pix | 0.5432 | 20.8420 | 0.7522 | 0.4243 | 0.3909 | 17.9528 | 0.6802 | 0.4909 |
| cGAN Fusion | 0.5976 | 21.5226 | 0.7713 | 0.3936 | 0.4220 | 17.8763 | 0.6897 | 0.4726 |
| DBPN | 0.5781 | 21.4716 | 0.7102 | 0.5101 | 0.4572 | 18.9330 | 0.6384 | 0.5910 |
| SRGAN | 0.5361 | 21.1968 | 0.6999 | 0.5261 | 0.4221 | 18.9772 | 0.6387 | 0.5694 |
| Ours (EAD) | 0.6470 | 22.4906 | **0.7904** | **0.3695** | 0.5225 | 19.7675 | **0.7280** | **0.4275** |
| Ours (EA64) | **0.6570** | **22.5552** | 0.7902 | 0.3764 | **0.5338** | **19.8547** | 0.7269 | 0.4342 |

by SSIM and FSIM), and obtain higher pixel accuracy (measured by PSNR) in both tasks compared to other approaches. With the more challenging task $t' < t$, where input HR images contain fewer constructed buildings than the ground truth HR images, our method exceeds the performance of other models by a greater margin.

Our results in Figure 3 confirm these findings with qualitative examples. Patches selected and zoomed in with colored bounding boxes show regions with significant change between $t$ and $t'$. Our model generates images with greater realism and higher structural information accuracy compared to baselines. While Pix2Pix and cGAN Fusion are also capable of synthesizing convincing images, they generate inconsistent building shapes, visual artifacts, and imaginary details like the swimming pool in the red bounding boxes. DBPN and SRGAN are faithful to information provided by the LR input but produce blurry images that are far from the ground truth.

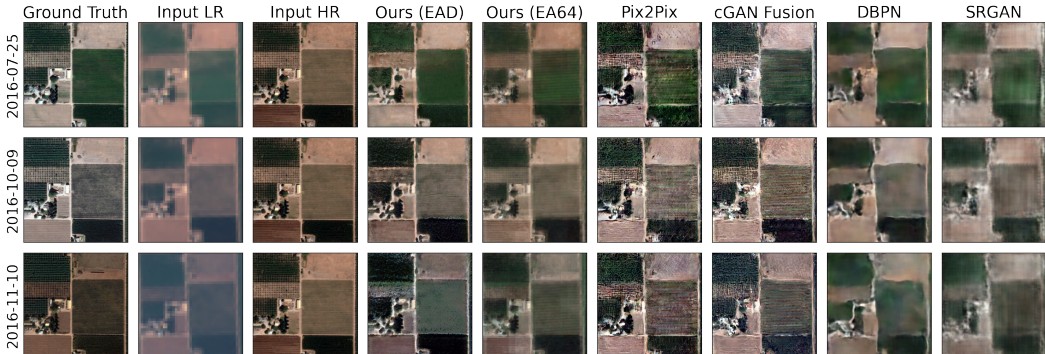

Figure 4: Samples from all models on the fMoW-Sentinel2 crop field dataset. Each row represents the results on the same location at a different timestamp given the same HR input from 2016-09-06.

**fMoW-Sentinel2 Crop Field Dataset**    We conduct experiments on the fMoW-Sentinel2 crop field dataset to compare model performance in settings with less data, fewer structural changes, and longer time series with unseen timestamps at test time. Our model outperforms baselines in all metrics, see Table 2. Figure 4 shows the image samples from different models on the fMoW-Sentinel2 crop field dataset. While image-to-image translation models fail to maintain structural similarity and SR models fail to attain realistic details, our model generates precise and realistic images.

**Discussion**    It is not surprising that our models generate high resolution details because they leverage a rich prior of what HR images look like, acquired via the cGAN loss, and GANs are capable of learning to generate high frequency details [19, 2]. Despite considerable information loss, inputs from LR devices still provide sufficient signal for HR image generation (e.g. swimming pools may change LR pixel values in a way that is detectable by our models but not by human perception). Experiments in Figure 3 and Section 5.5 show that these signals are enough for our model to reliably generate HR images that are high quality and applicable to downstream tasks.

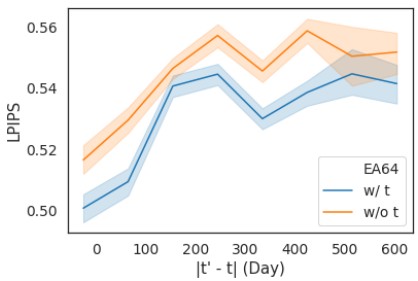

Figure 5: Ablation study on learning the temporal embeddings in our model using fMoW-Sentinel2 crop field dataset.

Table 2: Image sample quantitative results on fMoW-Sentinel2 crop field dataset.

| Model | SSIM↑ | PSNR↑ | FSIM↑ | LPIPS↓ |
|---|---|---|---|---|
| Pix2Pix | 0.2144 | 14.0276 | 0.6418 | 0.5847 |
| cGAN Fusion | 0.2057 | 14.1353 | 0.6409 | 0.5912 |
| DBPN | 0.3621 | 15.7878 | 0.6323 | 0.6428 |
| SRGAN | 0.3479 | 15.3502 | 0.6323 | 0.6301 |
| Ours (EAD) | 0.3526 | 16.5769 | **0.6887** | 0.5629 |
| Ours (EA64) | **0.3905** | **16.8879** | 0.6827 | **0.5197** |

In more extreme scenarios (e.g. LR captured by MODIS with 250m GSD v.s. HR captured by NAIP with 1m GSD), LR provides very limited information and therefore yields excessive uncertainty in generation. In this case, the high resolution details generated by our model are more likely to deviate from the ground truth.

It is worth noting that our LR images are captured by remote sensing devices (e.g. Sentinel-2), as opposed to synthetic LR images created by downsampling used in many SR benchmarks. As shown in our experiments, leading SR models such as DBPN and SRGAN do not perform well in this setting.

### 5.4 Ablation Study

We perform an ablation study on different components of our framework. We consider the following configurations for comparison: "No $G_P$" setting removes the pixel synthesizer to examine the effects of $G_P$; "Linear $F$" and "E only" use a single fully-connected layer and a single $3 \times 3$ convolutional layer with stride $= 1$ respectively to verify the influence of a deep multi-layer image feature mapper $F$. "ED Only" removes the global encoder $F_A$ and "A Only" removes the neighborhood encoder $F_E$ and decoder $F_D$. Note that because [2] has conducted thorough analysis on various settings of the positional encoder, we omit the configurations to assess the effects of spatial encoding in $E$.

As shown in Table 3, each component contributes significantly to performance in all evaluation aspects. While "EA64" outperforms "EAD" in SSIM and PSNR with a small margin, we observe slight checkerboard artifacts in the images generated by "EA64" (details in Appendix G). Overall, "EAD" is the most realistic to the human eye, which is consistent with the LPIPS results. However, "EA64" has stronger performance in a more data-constrained setting as shown in the fMoW-Sentinel2 crop field experiment. Samples generated by different configurations can be found in Appendix F.

We also demonstrate the effects of learning the time dimension in our model. Parameterizing our model with a continuous time dimension enables it to be applicable to time series of varying lengths with non-uniform time intervals (e.g. fMoW-Sentinel2 Crop Field dataset). Moreover, this parameterization also improves model performance. We train a modified version of "EA64" to exclude the time component in $E$ and compare the LPIPS values of the generated images to ones from the original "EA64" in Figure 5. In conjunction with the additional analysis in Appendix F, we show that the time-embedded model outperforms the same model without temporal encoding. Therefore, the time dimension is crucial to our model's performance, especially in areas with sparse HR satellite images over long periods of time.

We also include further details and analysis of our ablation study in Appendix F, including additional results for the effectiveness of the temporal dimension in $E$, comparison of different patch sizes for "EA" and training with different input choices.

### 5.5 Human Evaluation for Downstream Applications

Because our goal is to generate realistic and meaningful HR images that can benefit downstream tasks, we also conduct human evaluations to examine the potential of using our models for downstream applications. We deploy three human evaluation experiments on Amazon Mechanical Turk to measure

Table 3: Ablation study on the effects of different components of our model on Texas housing dataset. "+" represents adding certain components, "-" represents removing the components, and "*" represents different configurations from the original setting. See Section 5.4 for more details.

| Model | $F_E$ | $F_A$ | $F_D$ | $G_P$ | SSIM↑ | PSNR↑ | FSIM↑ | LPIPS↓ |
|---|---|---|---|---|---|---|---|---|
| "No $G_P$" | + | + | + | - | 0.5338 | 20.2712 | 0.7399 | 0.4482 |
| "Linear $F$" | * | - | - | + | 0.4585 | 18.8164 | 0.7006 | 0.4845 |
| "E Only" | * | - | - | + | 0.4761 | 19.0881 | 0.7146 | 0.4604 |
| "ED Only" | + | - | + | + | 0.5414 | 20.2488 | 0.7392 | 0.4340 |
| "A Only" | - | + | - | + | 0.5280 | 20.0312 | 0.7196 | 0.4418 |
| "EA64" | + | + | - | + | **0.5954** | **21.2050** | 0.7586 | 0.4053 |
| "EAD" | + | + | + | + | 0.5848 | 21.1291 | **0.7592** | **0.3985** |

Table 4: Human evaluation results on Texas housing dataset.

| Images | $r^2$ with mean count | | $r^2$ with median count | | % times selected | |
|---|---|---|---|---|---|---|
| | Buildings | Pools | Buildings | Pools | Similarity | Realism |
| HR $t'$ | 0.1475 | 0.1009 | 0.1595 | 0.1997 | - | - |
| DBPN | 0.8785 | 0.0227 | 0.8823 | -0.0640 | 1.75% | 1.25% |
| cGAN Fusion | 0.8793 | -0.0707 | 0.9093 | -0.0367 | 45.00% | 49.00% |
| Ours (EAD) | **0.9174** | **0.6158** | **0.9298** | **0.5953** | **53.25%** | **49.75%** |

the object reconstruction performance, similarity to ground truth HR images, and perceived realism of images generated by different models.

**Building and Swimming Pool Count** Object counting in HR satellite imagery has numerous applications, including environmental regulation [22], aid targeting [33], and local-level poverty mapping [3]. Therefore, we choose object counting as the primary downstream task for human evaluation. We randomly sample 200 locations in the test set of our Texas housing dataset, and assess the image quality generated under the setting $t' = 2018 > t = 2016$. Each image is evaluated by 3 workers, and each worker is asked to count the number of buildings as well as the number of swimming pools in the image. In each location, we select images generated from our model (EAD), cGAN Fusion, and DBPN, as well as the corresponding ground truth HR image $I_{hr}^{(t)}$ from 2016 and HR image $I_{hr}^{(t')}$ from 2018 (denoted as HR $t'$). We choose buildings and swimming pools as our target objects since both can serve as indicators of regional wealth [3, 8] and both occur with high frequency in the areas of interest. Swimming pools are particularly challenging to reconstruct due to their small size and high shape variation, making them an ideal candidate for measuring small-scale object reconstruction performance.

We measure the performance of each setting using the square of Pearson's correlation coefficient ($r^2$) between true and estimated counts, as in previous research [3]. As human-level object detection is still an open problem especially for satellite imagery [24, 34], human evaluation on this task serves as an upper bound on the performance of automatic methods on this task.

As shown in Table 4, our model outperforms baselines on both tasks, with the most significant performance advantage in the swimming pool counting task. Note that in rapidly changing environments like our Texas housing dataset, using the HR image of a nearby timestamp $t'$ cannot provide an accurate prediction of time $t$, which indicates the importance of obtaining higher temporal resolution in HR satellite imagery. Our model maintains the best object reconstruction performance among all models experimented, especially for small scale objects.

**Similarity to Ground Truth and Image Realism** Aside from object counting, we also conduct human evaluation on the image sample quality. With 400 randomly selected testing locations in our Texas housing test set, each worker is asked to either select the generated image that best matches a given ground truth HR image, or select the most realistic image from 3 generated images shown in random order. All images are generated under the setting $t' = 2018 > t = 2016$, and we choose the same models as the ones in the object counting experiment. Human evaluation results on image

sample quality align with our quantitative metric results. Our model produces the most realistic and accurate images among the compared models. Note that although cGAN Fusion generates realistic images, it fails to maintain structural information accuracy, resulting in lower performance in the similarity to ground truth task.

## 5.6 Temporal Extrapolation

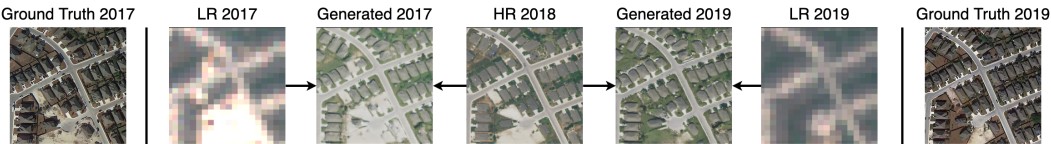

Figure 6: Temporal extrapolation application of our model. HR 2018 is the input NAIP image to both generated images shown in the figure. LR 2017 and LR 2019 are Sentinel-2 images of the same region in 2017 and 2019. Since NAIP imagery is not available in Texas for 2017 and 2019, the ground truth is obtained via Google Earth Pro. Note that the capture dates of the ground truth and the LR images are not perfectly aligned due to a lack of image availability.

Given a HR image at any time and LR images at the desired timestamps, we provide some evidence that our model is able to generate HR images at timestamps unavailable in the training dataset. Figure 6 demonstrates an example of such an application of our model. With the LR images from Sentinel-2, we generate HR images in 2017 and 2019, two years that do not have NAIP coverage in Texas. We compare the generated images with ground truth acquired from Google Earth Pro since the corresponding NAIP images are not available. Although the timestamps of the ground truth and LR images are not perfectly aligned, our generated images still show potential in reconstructing structural information reflected in the ground truth. More rigorous assessment of temporal extrapolation performance requires a more extensive dataset, which we leave to future work.

## 6 Conclusion and Statement of Broader Impact

We propose a conditional pixel synthesis model that uses the fine-grained spatial information in HR images and the abundant temporal availability of LR images to create the desired synthetic HR images of the target location and time. We show that our model achieves photorealistic sample quality and outperforms competing baselines on a crucial downstream task, object counting.

We hope that the ability to extend access to HR satellite imagery in areas with temporally sparse HR imagery availability will help narrow the data gap between regions of varying economic development and aid in decision making. Our method can also reduce the costs of acquiring HR imagery, making it cheaper to conduct social and economic studies over larger geographies and longer time scales.

That being said, our method does rely on trustworthy satellite imagery provided by reliable organizations. Just like most SR models, our model is vulnerable to misinformation (e.g. the failure cases presented in Appendix G due to unreliable LR input). Therefore, we caution against using generated images to inform individual policy (e.g. retroactively applying swimming pool permit fees) or military decisions. Exploration of performance robustness to adversarial examples is left to future study. Furthermore, we acknowledge that increasing the temporal availability of HR satellite imagery has potential applications in surveillance. Finally, we note that object counting performance is measured through human evaluation due to dataset limitations, and we leave measurement of automated object counting performance to future work.

## 7 Acknowledgement

This research was supported in part by NSF (#1651565, #1522054, #1733686), ONR (N00014-19-1-2145), AFOSR (FA9550-19-1-0024), ARO (W911NF-21-1-0125), Sloan Fellowship, HAI, IARPA, and Stanford DDI.

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
