# Spatial-Temporal Super-Resolution of Satellite Imagery via Conditional Pixel Synthesis

Yutong He     Dingjie Wang     Nicholas Lai     William Zhang     Chenlin Meng
Marshall Burke     David B. Lobell     Stefano Ermon
Stanford University
{kellyyhe, daviddw, nicklai, wxyz, chenlin, ermon}@cs.stanford.edu
{mburke, dlobell}@stanford.edu

## A    Model Details

We implement $F_E$ and $F_D$ with convolutional layers with kernel size $> 1$ with detailed specification below. All convolutional layers are followed by LeakyReLU activation.

$F_A$ is a self-attention module that takes $I_{ne}^{(t)}$ as the input and learns functions $Q, K$ : $\mathbf{R}^{C_{fea} \times H' \times W'} \to \mathbf{R}^{C_{fea}/8 \times H'W'}, V : \mathbf{R}^{C_{fea} \times H' \times W'} \to \mathbf{R}^{C_{fea} \times H'W'}$ and a scalar parameter $\gamma$ to map $I_{ne}^{(t)}$ to $I_{gl}^{(t)}$. Inspired by [9, 8], $I_{gl}^{(t)} = \gamma * v\text{softmax}(k^T q) + I_{ne}^{(t)}$, where $q = Q(I_{ne}^{(t)}), k = K(I_{ne}^{(t)}), v = V(I_{ne}^{(t)})$, Each entry $\beta_{(x',y'),(x,y)}$ in $k^T q$ indicates the extent of attention required from $(x, y)$ when extracting features for $(x', y')$. Note that $softmax(k^T q)v$ needs to be reshaped into $\mathbf{R}^{C_{fea} \times H' \times W'}$ before the addition.

We provide two versions of the image feature mapper. In version "EAD", we use two $3 \times 3$ convolutional layers with stride $= 2$ in $F_E$, a single self-attention module in $F_A$ and two $3 \times 3$ transposed convolutional layers with stride $= 2$ in $F_D$. In version "EA", we use one linear layer to map the channels from dimension $C$ to $C_{fea}$ and three $3 \times 3$ convolutional layers with stride $= 1$ in $F_E$, a single self-attention module in $F_A$ and an identity function in $F_D$. A skip connection is also added between the linear layer in $F_E$ and the output of $F_A$ in "EA".

Style injection is used in all fully-connected layers in $G_p$ and LeakyReLU is also applied after each fully-connected layer except for the layers that map to the output dimension.

### A.1    Training and Inference by Patch

Since HR image data can be high dimensional, it may not be feasible to process the entire $H \times W$ at once especially for the computationally intensive modules like matrix multiplication in the self-attention layer. Hence we utilize the benefit of the spatially bounded image coordinate grid and the conditional independent generation in $G_P$ to perform patch-based training and inference.

Without processing the original $H \times W$ image, the model instead takes a $H'' \times W''$ patch, whose coordinate grid can be represented as $P = \{(x, y, t) | h \le x \le h + H'', w \le y \le w + W''\}$ where $(h, w, t)$ is the top-left corner of the patch, as its input. $H''|H$ and $W''|W$. $F$ treats the patch as a complete image and calculates $I_{fea}^{(t)}(x, y)$ for $(x, y) \in P$. While $e_{f_o}$ still learns the Fourier feature for $(x, y, t)$, we grid sample the corresponding parameters in $e_{co}$ so that $e_{co}$ still maintains the original $H \times W$ dimension. Since the calculation in $G_P$ is conditionally independent among pixels given $I_{fea}^{(t)}$ and $E(x, y, t)$, the calculation of $G_P(E(x, y, t), F(I_{cat}^{(t)}), z)$ remains unchanged.

At training time, we randomly crop a $H'' \times W''$ patch as the input; at inference time, we use the same $z$ to generate all patches in the same image to maintain style consistency, and a sliding window technique described in Algorithm 1 to mitigate borderline artifacts.

35th Conference on Neural Information Processing Systems (NeurIPS 2021).

**Algorithm 1:** Sliding Window Generation

---

**Input:** G: Generator, S: patch size, H, W: size of the original image, $\lambda_s$: patch weight

**Result:** $I_{hr}^{(\hat{t})} \in \mathbf{R}^{C \times H \times W}$

$S_q = S/4$;

**for** *i in {0, ..., H/S-1}* **do**

    **for** *j in {0, ..., W/S-1}* **do**

        $I_{hr}^{(\hat{t})}(X) := G(X, z | I_{lr}^{(t)}, I_{hr}^{(t')})$ where
$X = \{(x,y) | iS \le x \le (i+1)S, jS \le y \le (j+1)S\}$

    **end**

**end**

**for** *i in {0, ..., H/S-1}* **do**

    **for** *j in {0, ..., W/S-2}* **do**

        $I_{temp}^{(\hat{t})}(X') = G(X', z | I_{lr}^{(t)}, I_{hr}^{(t')})$ where
$X' = \{(x,y) | iS \le x \le (i+1)S, (j+1)S - 2S_q \le y \le (j+1)S + 2S_q\}$

        $I_{hr}^{(\hat{t})}(X'') + = \lambda_s I_{temp}^{(\hat{t})}(X'')$ where
$X'' = \{(x,y) | iS \le x \le (i+1)S, (j+1)S - S_q \le y \le (j+1)S + S_q\}$

        $I_{hr}^{(\hat{t})}(X'') / = (1 + \lambda_s)$

    **end**

**end**

**for** *i in {0, ..., H/S-2}* **do**

    **for** *j in {0, ..., W/S-1}* **do**

        $I_{temp}^{(\hat{t})}(X') = G(X', z | I_{lr}^{(t)}, I_{hr}^{(t')})$ where
$X' = \{(x,y) | (i+1)S - 2S_q \le x \le (i+1)S + 2S_q, jS \le y \le (j+1)S\}$

        $I_{hr}^{(\hat{t})}(X'') + = \lambda_s I_{temp}^{(\hat{t})}(X'')$ where
$X'' = \{(x,y) | (i+1)S - S_q \le x \le (i+1)S + S_q, jS \le y \le (j+1)S\}$

        $I_{hr}^{(\hat{t})}(X'') / = (1 + \lambda_s)$

    **end**

**end**

---

## B  Implementation Details

We choose $H = W = 256$, $C = 3$ (the concatenated RGB bands of the input images), $C_{fea} = 256$, $m = 3$, $n = 14$ and and $\lambda = 100$ for all of our experiments. We use non-saturating conditional GAN loss for $G$ and $R_1$ penalty for $D$, which has the same network structure as the discriminator in [7, 1]. We train all models using Adam optimizer with learning rate $2 \times 10^{-3}$, $\beta_0 = 0, \beta_1 = 0.99, \epsilon = 10^{-8}$ on NVIDIA Titan XP GPUs. We train each model to convergence which takes around 4-5 days on 1 NVIDIA Titan XP GPU. EAD models can sample 10 images per second and EA64 models can sample 1500 images in around 19 minutes (1.3 image/s). The difference in inference times results from the inference by patch technique described in Appendix A.1.

In our Texas housing experiment, we use time unit $u = 2$ and denote the images from 2016 as $t = 0$ and the images from 2018 as $t = 2$. In our fMoW-Sentinel2 crop field experiment, we use $u = 365$ and denote the starting date of Sentinel-2 imagery, 2015-06-23, as $t = 0$. The temporal coordinates are then normalized with chosen time unit. We implement our code based on the official PyTorch implementation of CIPS [1].

We use the same backbone network structures for cGAN Fusion and Pix2Pix with 6-channel input $I_{cat}^{(t)}$. Because neither SRGAN nor DBPN provides $10 \times$ SR, we resize the LR images to $32 \times 32$ and perform $8 \times$ enlargement.

Pixel values are re-scaled to $[0, 1]$ during evaluation. We use [4]'s implementation for SSIM, FSIM and LPIPS, and implement PSNR based on [2]'s implementation.

# C Datasets

## C.1 Texas Housing Dataset

We collect geo-coordinates for residential houses effectively built between 2014 and 2017 in Texas from the CoreLogic tax and deed database. To make data extraction more efficient, we use DBSCAN [5] to spatially cluster the geo-locations of the houses, and choose four clusters around major metropolitan areas in Texas (Austin/San Antonio, Dallas, Houston, and Waco) which encapsulate most of the datapoints obtained from the CoreLogic database. We then find the latitude and longitude coordinates that bound each cluster and extract rectangles for both NAIP and Sentinel-2 from Google Earth Engine (GEE) [6] so we can run fast local extraction. The Sentinel-2 rectangles are cloud filtered using the Sentinel-2 Cloud Probability image collection[1], and the NAIP rectangles are cloud filtered by its data collection process. We choose to obtain images from year 2016 and 2018, which is the range of mutual availability of NAIP and Sentinel-2 in Texas. NAIP coverage of Texas ranges from August to September for both years, so we acquire the corresponding Sentinel-2 image from the same time range.

We then extract 4 images for each house (NAIP in 2016, NAIP in 2018, Sentinel-2 in 2016, and Sentinel-2 in 2018). NAIP images have a resolution of 1 meter per pixel, Sentinel-2 images are 10 meters per pixel, and we acquire the RGB bands from both devices. Our dataset consists of 286717 houses and their surrounding neighborhoods. We reserve 14101 houses from 20 randomly selected zip codes as the testing set and use the remaining 272616 houses from the other 759 zip codes as the training set. In total, there are 1146868 multi-resolution images collected from different sensors for our experiment. Table 1 shows a distribution of houses by year and also by region. Figure 1 shows an example of a house captured using this process with NAIP and Sentinel-2.

For both NAIP and Sentinel-2 images, we export from GEE's image pyramid at scale 1, which is equivalent to applying geo-referenced nearest neighbor resampling to the Sentinel-2 images in order to obtain images with the same dimensions as NAIP. Each neighborhood has radius of 0.001 degrees. The resulting images from both devices have dimensions of around $256 \times 256$.

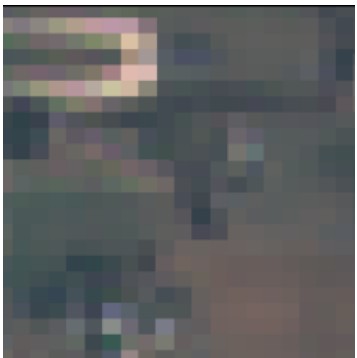 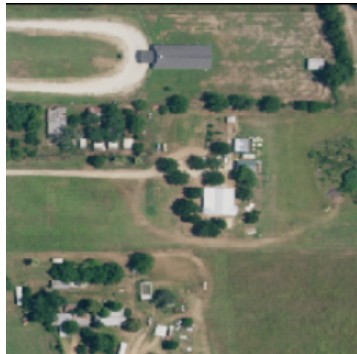

(a) Low resolution image (Sentinel-2)    (b) High resolution image (NAIP)

Figure 1: Sentinel-2 and NAIP image pair example.

Table 1: Distribution of collected houses by year and region.

| Years | Austin | Dallas | Houston | Waco | **Total** |
|-------|--------|--------|---------|------|-----------|
| 2014 | 19630 | 24625 | 20327 | 2035 | **66617** |
| 2015 | 21673 | 27797 | 20347 | 2100 | **71917** |
| 2016 | 20327 | 30329 | 17741 | 2299 | **70696** |
| 2017 | 23997 | 32777 | 18312 | 2401 | **77487** |
| **Total** | **85627** | **115528** | **76727** | **8835** | **286717** |

Sentinel-2 data is provided courtesy of "Copernicus Sentinel data 2015-2020" as outlined by the European Space Agency (ESA), and NAIP data is provided courtesy of U.S. Department of Agriculture

---

[1]Guidance for cloud filtering using the Sentinel-2 Cloud Probability dataset is available here.

(USDA) Farm Production and Conservation - Business Center, Geospatial Enterprise Operations. We have removed all personally identifiable information from our curated dataset.

## C.2 fMoW-Sentinel2 Crop Field Dataset

We derive this dataset from the crop field category of Functional Map of the World (fMoW) dataset [3]. We take RGB images from the fMoW crop field object category due to a high likelihood of changes over time compared to other object classes in the fMoW dataset. We pair each fMoW image (0.3m to 1m GSD) with a corresponding lower resolution Sentinel-2 RGB image (10m GSD) captured at the same location and within a 20 day range centered around the fMoW image capture time. We select the least cloudy image from available images within the 20 day capture range using the Sentinel-2 Cloud Probability dataset[2]. We prune locations with fewer than 2 timestamps, yielding 1752 locations and a total of 4898 fMoW-Sentinel2 pairs. Each location contains between 2-15 timestamps spanning from 2015 to 2017. We reserve 237 locations as the testing set and the remaining 1515 locations as the training set. We resize images to $256 \times 256$ using nearest neighbour interpolation during preprocessing. Figure 2 shows an example of an image pair after data collection and preprocessing.

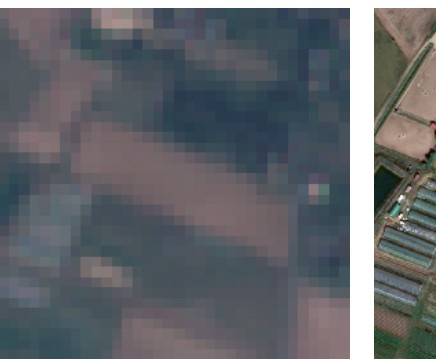 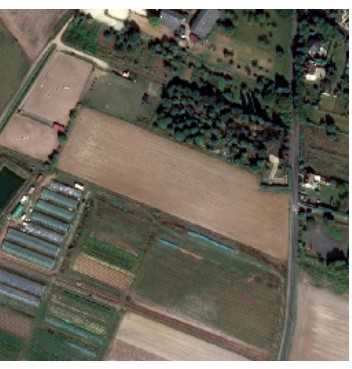

(a) Low resolution image (Sentinel-2)    (b) High resolution image (fMoW)

Figure 2: Sentinel-2 and fMoW image pair example.

Sentinel-2 data is provided courtesy of "Copernicus Sentinel data 2015-2020" as outlined by the European Space Agency (ESA), and fMoW images are used under the "Functional Map of the World Challenge Public License"[3]. The dataset does not contain personally identifiable information.

---

[2]The Sentinel-2 Cloud Probability dataset can be found here.

[3]The Functional Map of the World Challenge Public License can be found here.

# D   Human Evaluation

We categorize models into three groups: image-to-image translation models, super-resolution models and our models. For human evaluation experiments, we primarily compare cGAN Fusion, DBPN and our model (EAD) because they achieve the best quantitative metrics in each model category. We conduct all human evaluation under the setting of $t' > t$ on the Texas housing dataset.

## D.1   Building and Swimming Pool Count

We measure the object reconstruction performance of images at two scales through a task where workers count the number of buildings and swimming pools in a satellite image.

**Experiment setup**    200 locations are randomly sampled from the test set, and the corresponding images are collected from the following datasets/models: ground truth HR image $I_{hr}^{(t)}$ from 2016, HR image $I_{hr}^{(t')}$ from 2018 (HR $t'$), cGAN Fusion, DBPN and our model (EAD). 10 images and one vigilance test image are packed into each Human Intelligence Task (HIT) and 3 assignments are requested for each HIT, allowing for comparison of results from different workers.

**Instructions full text**    "Please only participate in this HIT if you have normal color vision. The HIT should take approximately 5 minutes to complete. There are 11 trials in the HIT, and each should take 15-30 seconds. You are shown a satellite image that may be real or computer generated. Please count the number of buildings and the number of swimming pools in the image. Please only count buildings and swimming pools that are fully contained within the image, not noticeably clipped off by the edges of the image. Objects that are just barely clipped off (less than 5%) may be counted. Objects in generated satellite images may look ambiguous. Give your best guess count in these cases. This can be subjective so follow your instincts! You will complete a short practice session (approximately 1 minute) before starting the main task."

**Vigilance Tests**    Each HIT contains a vigilance image for which object counts are sufficiently unambiguous. HITs that fail the vigilance test are automatically rejected.

**Worker qualification**    Workers are required to have no less than 97% HIT approval rate and no less than 5000 total approved HITs to qualify for our experiment. Worker uniqueness is used to prevent the same workers from completing more than 3 total HITs for this experiment.

**Compensation**    Workers are awarded $0.80 upon completion of a HIT, which has an expected completion time of 6 minutes, yielding an estimated hourly wage of $8. We deploy a total of 300 HITs, resulting in a total cost of $240 for this experiment, not including additional fees.

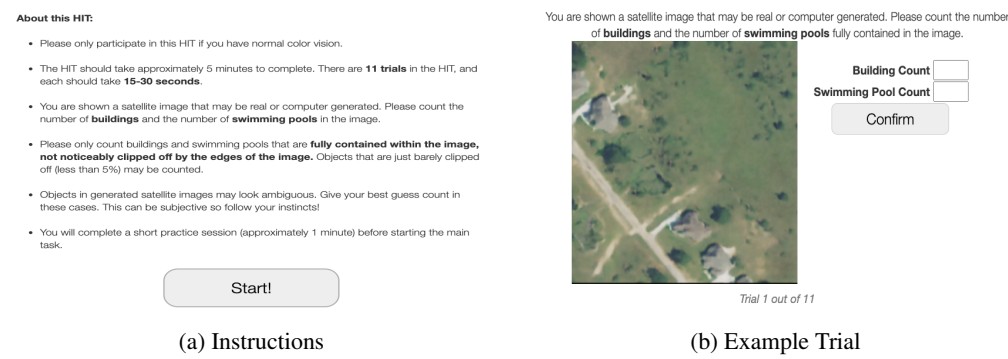

(a) Instructions                 (b) Example Trial

Figure 3: Instructions and example trial for the building and swimming pool count task.

## D.2 Similarity to Ground Truth

We measure both structural information accuracy and perceived realism of generated images through a task in which workers select the generated satellite image they observe to best match a real ground truth image.

**Experiment setup** 400 locations are randomly sampled from Texas housing dataset test set, and the corresponding images are collected from the following datasets/models: ground truth HR image $I_{hr}^{(t)}$ from 2016, cGAN Fusion, DBPN and our model (EAD). 20 sets of images and two sets of vigilance test images are packed into each HIT and 1 assignment is requested for each HIT.

**Instructions full text** "Please only participate in this HIT if you have normal color vision. The HIT should take 6-7 minutes to complete. There are 22 trials in the HIT, and each should take 12-15 seconds. You are shown a real satellite image and three computer generated satellite images. Please select the generated image that you feel best matches the real image based on image accuracy and realism. Image accuracy refers to the degree to which ground level structural information (e.g. buildings, trees, roads) matches the real image. Image realism refers to the degree that the generated image looks like a real satellite image. You will complete a short practice session (less than 1 minute) before starting the main task."

**Vigilance Tests** Each HIT contains a similarity trial for which the best matching image is sufficiently unambiguous. HITs that fail the vigilance test are automatically rejected.

**Worker qualification** Workers are required to have no less than 97% HIT approval rate and no less than 5000 total approved HITs to qualify for our experiment. Worker uniqueness is used to prevent the same workers from completing more than 1 HIT for this experiment.

**Compensation** Workers are awarded $1.00 upon completion of a HIT, which has an expected completion time of 6 minutes, yielding an estimated hourly wage of $10. We deploy a total of 20 HITs, resulting in a total cost of $20 for this experiment, not including additional fees.

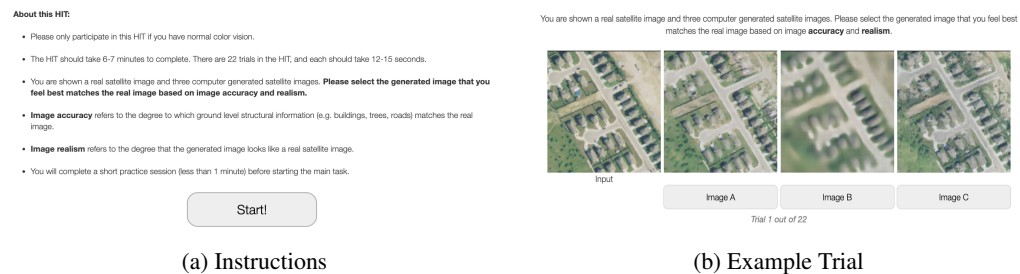

(a) Instructions          (b) Example Trial

Figure 4: Instructions and example trial for the similarity to ground truth task.

## D.3 Image Realism

We measure the perceived realism of generated images through a task in which workers select the generated satellite image they observe to look the most realistic. Workers are provided with real satellite images to examine in the beginning of the task.

**Experiment setup** 400 locations are randomly sampled from Texas housing test set, and the corresponding images are collected from the following models: cGAN Fusion, DBPN and our model (EAD). 20 sets of images and two sets of vigilance test images are packed into each HIT and 1 assignment is requested for each HIT.

**Instructions full text** "Please only participate in this HIT if you have normal color vision. The HIT should take 6-7 minutes to complete. There are 22 trials in the HIT, and each should take 15 seconds. You are shown three computer generated satellite images. Please select the generated image

that most resembles a high quality real satellite image. Please pay attention to the details of objects. You will complete a short practice session (approximately 1 minute) before starting the main task."

**Vigilance Tests**    Each HIT contains a realism trial for which the most realistic satellite image is sufficiently unambiguous. HITs that fail the vigilance test are automatically rejected.

**Worker qualification**    Workers are required to have no less than 97% HIT approval rate and no less than 5000 total approved HITs to qualify for our experiment. Worker uniqueness is used to prevent the same workers from completing more than 1 HIT for this experiment.

**Compensation**    Workers are awarded $1.00 upon completion of a HIT, which has an expected completion time of 6 minutes, yielding an estimated hourly wage of $10. We deploy a total of 20 HITs, resulting in a total cost of $20 for this experiment, not including additional fees.

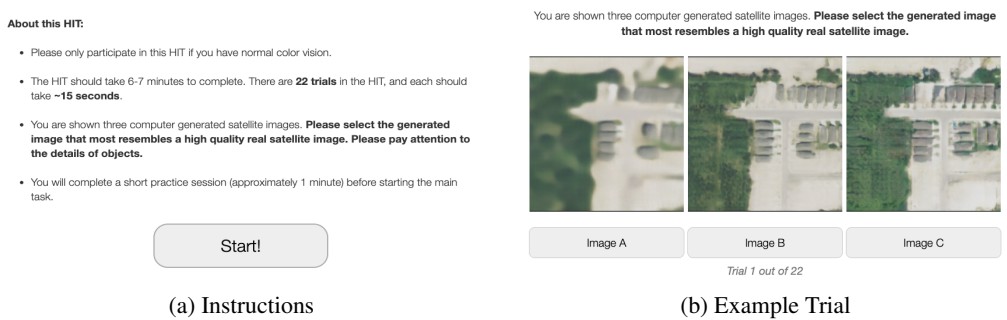

(a) Instructions                                    (b) Example Trial

Figure 5: Instructions and example trial for the image realism task.

## D.4    Additional Statistical Results

Figure 6 shows box plots of differences between mean object counts in each location for images generated by each model compared to the ground truth images. We use $IQR = Q_3 - Q_1$ and plot non-outliers in $[Q_1 - 1.5 \times IQR, Q_3 + 1.5 \times IQR]$. We remind the reader that object counts for ground truth images are also generated by human workers through the same experiment. Compared to our model, cGAN Fusion yields greater variation in object count difference and has a tendency to hallucinate swimming pools. Images generated by DBPN often yield under-counted buildings and swimming pools, likely due to image blurriness and signal limitations from LR input. We note that building counts obtained from the HR image at a future time $t'$ are frequently higher than building counts obtained from the HR image at a past time $t$, sometimes significantly so. This can be attributed to the rapidly changing environments and the presence of new house constructions in our dataset. Count differences derived from images generated by our model exhibit smaller variance and less data skew when measured against counts derived from the ground truth images.

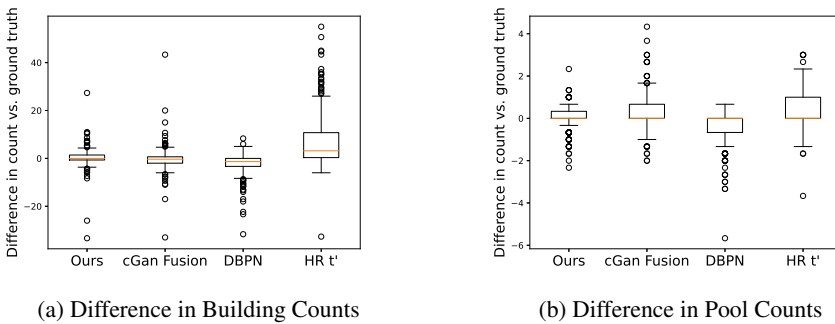

(a) Difference in Building Counts                (b) Difference in Pool Counts

Figure 6: Box plots of differences in mean object count for each dataset/model compared to the mean object counts yielded by the ground truth images.

## E    Additional Generation Results

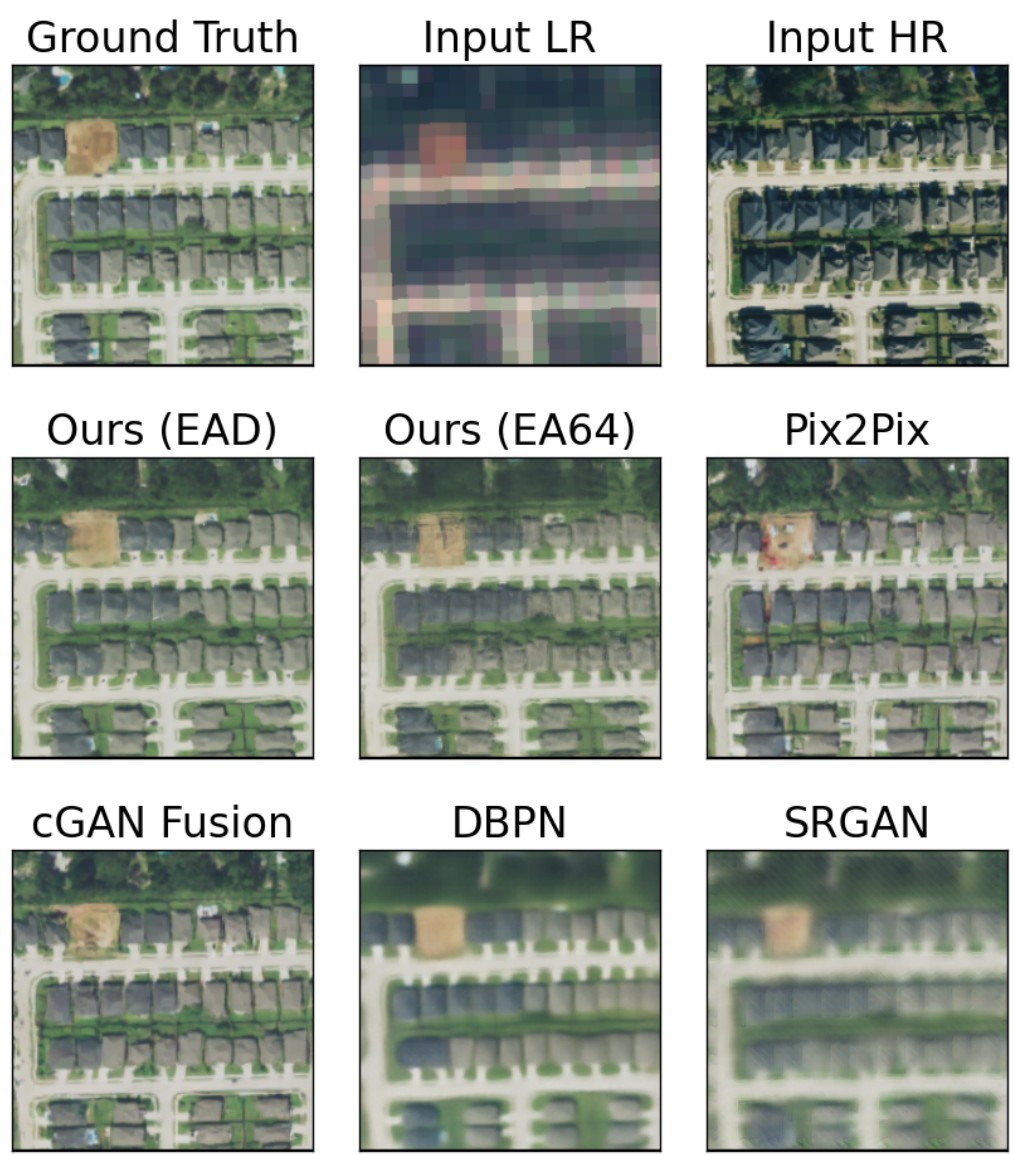

Figure 7: Additional samples from all models on the Texas housing dataset with setting $t' > t$.

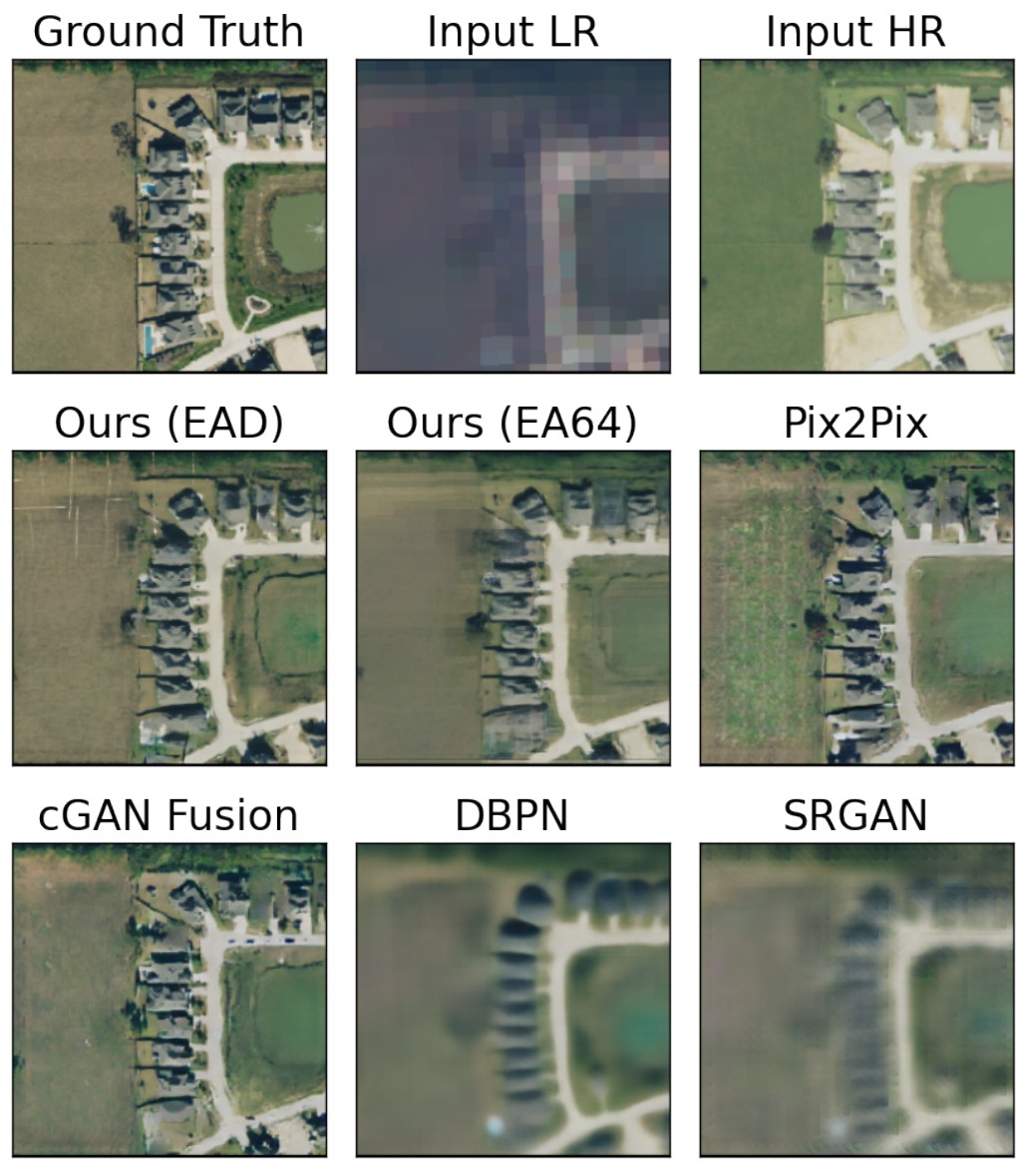

Figure 8: Additional samples from all models on the Texas housing dataset with setting $t' < t$.

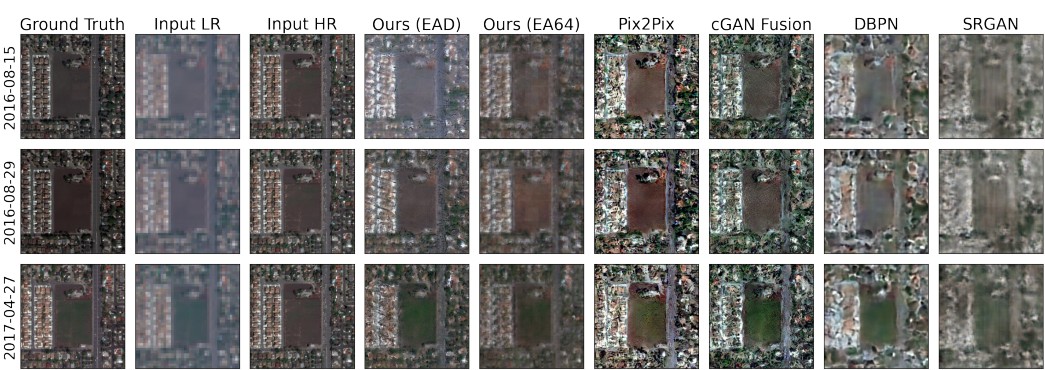

Figure 9: Additional samples from all models on the fMoW-Sentinel2 crop field dataset.

# F  Additional Ablation Study

In this section, we perform an additional ablation study with qualitative examples and quantitative analysis.

**Model Configurations**  Figure 11 shows an example of samples from different configurations of our model mentioned in Section 5.4. From the samples in Figure 11, we observe that although our model still produces convincing results without $G_P$, it suffers from the same high saturation visual artifacts as Pix2Pix and cGAN Fusion. Without deep network structures in $F$, while faithful to the LR image, our model fails to generate realistic results. Without $F_E$ and $F_D$, our model shows obvious checkerboard artifacts. After removing $F_A$, our model generates inconsistent shapes for objects in the image.

**Temporal Encoding**  We conduct an ablation study on learning the time dimension in $E$ using "EA64" on the fMoW-Sentinel2 crop field dataset. Figure 10 demonstrates a comparison of all metrics between models with and without learning the time component in $E$. We group all possible values of $|t' - t|$ in the test set with bin width of 90 days, and plot the mean of each bin and error bars with 75% confidence level.

As we observe in the figure, the performance of EA64 with time embeddings shows advantages in most metrics as the time difference between capture dates of the target and reference HR images increases, which suggests benefits in real life settings where HR imagery is often captured with long visiting cycles.

Using permutation test with differences between means to test the statistical significance, we verify our observation with the null hypothesis $H_0 : f(X_w) \leq f(X_{wo})$ where $f$ is a choice of metrics where higher values indicate better models. Here $f$ can be SSIM, PSNR, FSIM or -LPIPS compared to the ground truth. $X_w$ and $X_{wo}$ represent the generated images with and without learning the time dimension respectively. FSIM and LPIPS achieve p-values 0.002 and 0 respectively for the null hypothesis. PSNR achieves p-value=0.09 when |t'-t|>300 days, which shows that the advantage of the time-embedded model increases when the difference between $t'$ and $t$ increases.

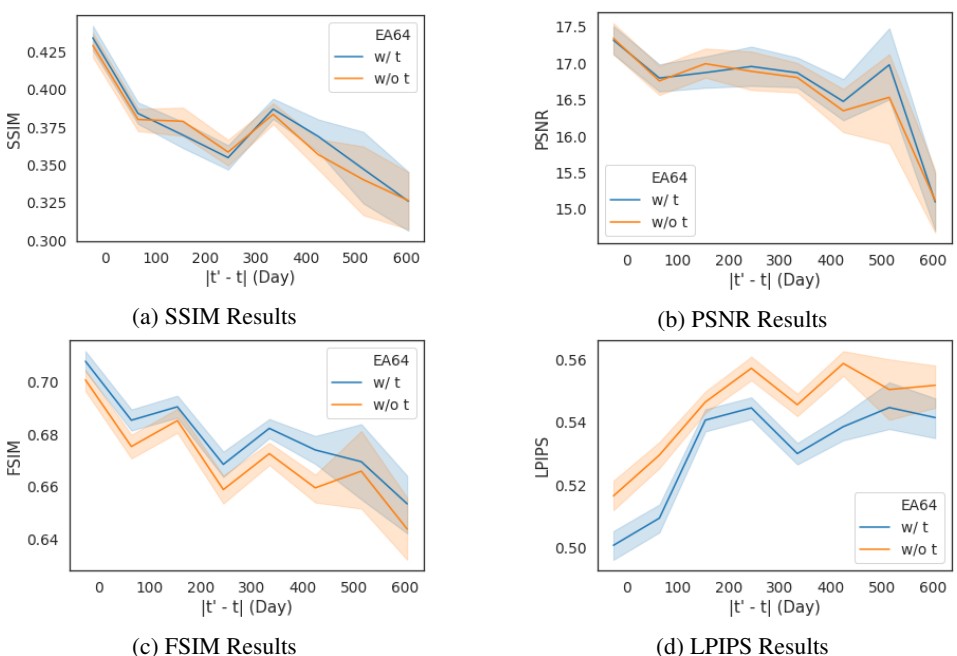

(a) SSIM Results

(b) PSNR Results

(c) FSIM Results

(d) LPIPS Results

Figure 10: Ablation study on learning the time dimension in our model (EA64) using fMoW-Sentinel2 crop field dataset.

Table 2: Quantitative results of ablation study on learning the temporal information on the fMoW-Sentinel2 crop field dataset. "w/ t" represent EA64 model with time dimension in $E$, and "w/o t" represent the same model without time dimension in $E$. As shown here, adding the temporal dimension substantially benefits model performance.

| Model | SSIM↑ | PSNR↑ | FSIM↑ | LPIPS↓ |
|-------|-------|-------|-------|--------|
| w/o t | 0.3844 | 16.8075 | 0.6774 | 0.5380 |
| w/ t | **0.3905** | **16.8879** | **0.6827** | **0.5197** |

We also report average quantitative results for both models and observe that the model with temporal information outperforms in all metrics on the fMoW-Sentinel2 crop field dataset. Hence we can conclude that learning the time dimension is crucial for our model performance.

**Patch Size**   We also investigate a setting of "EA" with patch size 32 denoted as "EA32", with quantitative results presented in Table 3. The model performance is degraded due to the reduction of the global view of $F_A$. We hypothesize that increasing the patch size can benefit the performance. However, further study is required as it is not feasible to train patch sizes that are significantly larger than 64 on our devices.

Table 3: Ablation study on the patch sizes for patch-based generation on the Texas housing dataset. $+(S)$ denotes using size $S$ patches during training and inference.

| Model | $F_E$ | $F_A$ | $F_D$ | $G_P$ | SSIM↑ | PSNR↑ | FSIM↑ | LPIPS↓ |
|-------|-------|-------|-------|-------|-------|-------|-------|--------|
| EA32 | +(32) | +(32) | - | + | 0.5767 | 20.8409 | 0.7480 | 0.4254 |
| EA64 | +(64) | +(64) | - | + | **0.5954** | **21.2050** | **0.7586** | **0.4053** |

**Input Source**   Here we provide an experiment using our model with (1) two additional LR image inputs from different time steps (Sentinel-2 images from 2017 and 2019) and (2) no HR reference input (as in standard SR approaches) in the Texas housing dataset.

We do not observe improvements from the "Multiple LR" setting, which agrees with our claim that given a LR image at the target time, other LR views from different time steps (in the past or future) provide little or no additional information. As shown for the "No HR t'" setting in the table below, our approach also has clear advantages over the standard SR setting (with no HR reference input). It is also worth noting that the LR images in our experiments are from real LR devices, which is different from synthetic LR images created by downsampling used in many SR benchmarks. Leading standard SR methods such as DBPN and SRGAN do not perform well as shown in the experiments in Section 5.

Table 4: Ablation study on different input choices on the Texas housing dataset. "Multiple LR" represents the setting that uses more than one LR images from different time steps as inputs, and "No HR t'" represents the setting that uses no HR reference image from a different time and only takes the LR image as the input, as in the standard SR approaches. Our approach outperforms both models with the EAD configuration.

| Model | t' > t | | | | t' < t | | | |
|-------|--------|-------|-------|--------|--------|-------|-------|--------|
| | SSIM↑ | PSNR↑ | FSIM↑ | LPIPS↓ | SSIM↑ | PSNR↑ | FSIM↑ | LPIPS↓ |
| Multiple LR | 0.6266 | 22.0939 | 0.7814 | 0.3873 | 0.5023 | 19.5255 | 0.7198 | 0.4428 |
| No HR t' | 0.4731 | 19.9537 | 0.7047 | 0.4991 | 0.3535 | 17.4470 | 0.6453 | 0.5400 |
| Ours | **0.6470** | **22.4906** | **0.7904** | **0.3695** | **0.5225** | **19.7675** | **0.7280** | **0.4275** |

Our method can also be easily extended to include additional bands such as NIR by changing the number of input channels. We choose RGB bands in this work because they are commonly available in remote sensing devices and they are sufficient for our target tasks. Thus we see our results as a lower bound of what can be achieved, and enhancements using additional bands are left to future study.

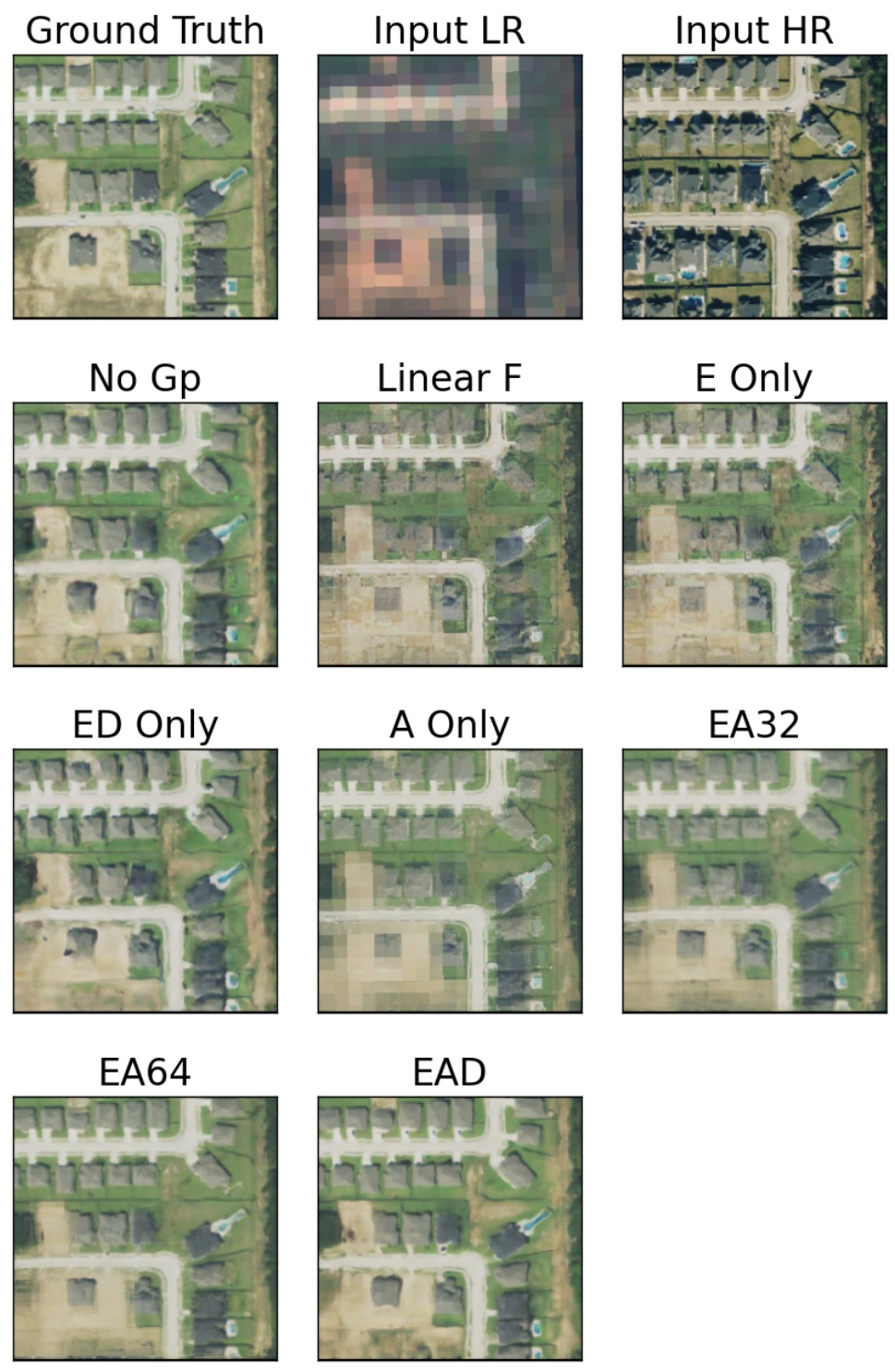

Figure 11: Samples from all different configurations of our model on the Texas housing dataset with setting $t' > t$.

## G   Failure Cases

In this section, we analyze failure cases of our model. Figure 12 is an example where we observe unsatisfactory generation quality from our model. In some edge cases, such as extreme snow reflection in the LR input image, our model is unable to successfully reconstruct an accurate image of the ground truth. In this case, our EAD setting does not obtain sufficient color accuracy. Our EA64 setting is able to generate more accurate color details for the 2017-01-05 image, but it synthesizes obvious checkerboard artifacts and is therefore less realistic to human perception.

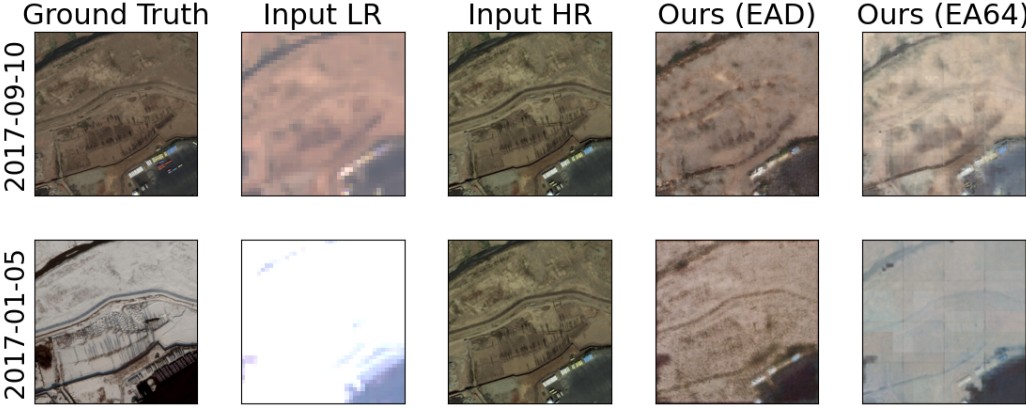

Figure 12: Failure cases when using fMoW-Sentinel2 crop field dataset with input HR captured on 2016-09-29. With this edge case LR input (extreme snow reflection), our EAD model fails to generate accurate color and our EA64 model exhibits obvious checkerboard artifacts.