# OpenReview forum: "Spatial-Temporal Super-Resolution of Satellite Imagery via Conditional Pixel Synthesis"
_NeurIPS.cc/2021/Conference — NeurIPS 2021 Poster_

### Official Review · Reviewer_F5jU · 2021-07-14

**Rating:** 7
**Confidence:** 4

**Summary:**

The authors propose a conditional pixel synthesis approach which combines frequent low-resolution aerial imagery with less widely available high-resolution imagery to generate photo-realistic and accurate high-resolution imagery.  The approach is largely based on NeRF with modifications for the remote sensing domain.  Results are compared to other techniques such as image fusion models, Pix2Pix, SuperResolution GAN, and others.

**Limitations And Societal Impact:**

In the Implementation Details I see the hardware used, but do not see any indication as to the time trained (either in epochs/steps, or wall time).  Similarly, there is no mention to inference speed or limitations.  Because this paper is focused on a domain involving huge amounts of data, I believe some discussion around this is warranted.  For example, the authors cite an important usecase of the HR data as exploring poverty metrics.  However, if the cost to run such a model is massive, perhaps there is some drawback to this approach.

Societal Impacts, particularly in reference to security/surveillance are mentioned, but not discussed at length.  There are some fairly significant implications here both in terms of individual privacy as well as defense/military applications.  Especially regarding the later, generating high-resolution images with military implications could be problematic if the model is incorrect (both in terms of false positives as well as negatives).  Photorealistic images could be generated and shared with the public to bolster support for political agendas- I believe speaking to this even briefly is an important consideration.

**Main Review:**

The authors address an important issue in remote sensing which is the availability of high-resolution imagery.  Frequent, low-resolution imagery is abundant, but high-resolution imagery is much more difficult to obtain.  They use the combination of LR and HR data to produce a photorealistic HR data at the point of interest.  They demonstrate this is both photo-realistic as well as useful for downstream tasks such as counting.

The approach is largely based on NeRF although addresses challenges specific to the remote sensing domain, such as the construction of the 3D scene.  The theoretical derivation is clear, appears correct, and is easy to follow.  However, it would be useful to highlight where the current approach differs from some of the cited related works as NeRF and its derivatives are a relatively new approach.  It is hard to tell which elements are completely new, adapted, or directly used from these other works.
 The authors reference how their approach differs in relation to SR and Fusion RM approaches, but doesn't draw a direct comparison to what modifications are made relative to the works cited around NeRF and Pixel Synthesis.  Within the derivation, it would be useful to include remarks such as "following the approach of" or "unlike X which did Y", to highlight the originality of the various components in their approach.

Experiments are performed on two datasets with different domains- Texas Housing Dataset which is more "industrial" focused, and Sentinel2 Crop Field Dataset, which has a more "agricultural focus".  This is a very nice addition as it shows the approach is applicable over two domains with likely different statistics (THD likely containing more geometric features than Sentinel2).  Performing a downstream counting task is also powerful in addition to providing a human-perception evaluation.  The only piece I would be interested in seeing is a similar task on the agriculture dataset- there are agricultural metrics and tasks which are often very color-focused as opposed to shape focused.  Even if the images look photo-realistic, slight color variations could be problematic on downstream tasks such as determining crop type, land usage, growth stage, stress/damage, etc.  Those tasks may rely on subtle qualities of the image which the human does not perceive easily.  Performing a similar downstream task on this dataset would be very useful.

Ablation studies are performed to determine the impact on the different components of the framework.

Regarding both the experiments and ablation studies, while the improvements are so large that it is reasonable to believe they are significantly better there is an increasing desire to see experiments run multiple times so that errorbars around these measurements can be provided and the degree of significance can be determined.

There is always a worry around adversarial examples, especially in a generative setting, especially when there may be surveillance use cases.  It would be interesting to see how small pixel shifts, particularly in the low-resolution imagery, impacts the final generated results.

The paper is overall clear and well written.  Implementation details are given in the appendix as opposed to the main text, which is suboptimal.  Because all elements enabling reproducibility should be included in the main text, I believe it's important to move as much of that as possible to the body of the text.



**Time Spent Reviewing:**

3

---

> ### Author Response · Authors · 2021-08-10
> **Additional details and discussions based on the suggestions**
>
> Thank you for your comprehensive review! We appreciate your constructive comments.
> 1. **Comparison with Other NeRF-like models:** We have included comparison between our model and other NeRF/Pixel Synthesis models in LL 77-86. We will add the corresponding remarks to emphasize the originality of our work.
> In summary, our model is novel in that: it is a pixel synthesis model that learns a conditional 2D spatial coordinate grid along with a continuous time dimension, which is tailored for remote sensing where the same location/object can be captured by different devices (eg. NAIP or Sentinel-2) at different times (eg. year 2016 or 2018).
> 2. **Downstream Tasks on fMoW-Sentinel2 Crop Field Dataset:** We are very interested in applying our model on downstream tasks on an agricultural dataset. However, since we lack ground truth labels for crop types, land usage, growth stage and stress/damage, etc, it is difficult for fMoW to be directly applicable to those tasks. We would love to explore other datasets with corresponding labels in the future.
> 3. **Error bars:** Here we provide the mean and std for 5 runs of evaluations on fMoW-Sentinel2 Crop Field dataset. Our conclusion does not change according to this result, and we will add the error bars for multiple training runs in our paper as well.
> |    Model    	|        SSIM        	|         PSNR        	|        FSIM       	|       LPIPS       	|
> |:-----------:	|:------------------:	|:-------------------:	|:-----------------:	|:-----------------:	|
> |   Pix2Pix   	|  0.2141$\pm$1.71$\times$10$^{-4}$  	|  14.0170$\pm$4.40$\times$10$^{-3}$  	|  0.6415$\pm$1.05$\times$10$^{-4}$ 	|  0.5848$\pm$1.09$\times$10$^{-4}$ 	|
> | cGAN Fusion 	|  0.2058$\pm$8.700$^{-5}$ 	| 14.1394$\pm$3.04$\times$10$^{-3}$  	| 0.6409$\pm$3.76$\times$10$^{-5}$ 	| 0.5913$\pm$7.87$\times$10$^{-5}$ 	|
> |     DBPN    	|    0.3621$\pm$0.00   	|    15.7878$\pm$0.00   	|   0.6323$\pm$0.00   	|   0.6428$\pm$0.00   	|
> |    SRGAN    	|    0.3479$\pm$0.00   	|    15.3502$\pm$0.00   	|   0.6323$\pm$0.00   	|   0.6301$\pm$0.00   	|
> |     EAD     	| 0.3523$\pm$2.02$\times$10$^{-4}$  	|  16.5773$\pm$3.57$\times$10$^{-3}$ 	| 0.6888$\pm$6.68$\times$10$^{-5}$ 	| 0.5630$\pm$9.43$\times$10$^{-5}$ 	|
> |     EA64    	|  0.3904$\pm$2.24$\times$10$^{-5}$ 	|  16.8907$\pm$1.74$\times$10$^{-3}$ 	| 0.6827$\pm$3.27$\times$10$^{-5}$ 	| 0.5197$\pm$3.90$\times$10$^{-5}$ 	|
> 4. **Adversarial Examples:** Our models do rely on trustworthy satellite imagery provided by reliable organizations. Just like most SR models, our model is vulnerable to these attacks (e.g. the failure cases presented in Appendix G due to unreliable LR input). Thus we suggest using our method with truthful datasets and caution against using our model as the only method to inform individual policy decisions (e.g. retroactively applying swimming pool permit fees). It is an interesting future direction to see how vulnerable our model is to these attacks and how we can make it more robust.
> 5. **Implementation Details:** We train each model to convergence which takes around 4-5 days on 1 NVIDIA Titan XP GPU. EAD can sample 1500 images in around 2.5 minutes (10 images/s) and EA64 can sample 1500 images in around 19 minutes (1.3 image/s). The difference in inference times results from the inference by patch technique described in Appendix A.1.
> 6. **Potential Negative Societal Impact:** Due to the vulnerability discussed in **Adversarial Examples**, we caution against using generated images as the only source to inform individual policy or defense/military decisions. Additionally, we acknowledge that certain areas of the globe may still have no high resolution satellite coverage [1], in which case our approach may exacerbate the data gap by only augmenting data availability in regions that already have some HR satellite imagery.
>
> **Reference:**
> [1] Reid, Elizabeth. “A Look Back at 15 Years of Mapping the World.” Google, Google, 6 Feb. 2020, blog.google/products/maps/look-back-15-years-mapping-world/.

---

### Official Review · Reviewer_Gndk · 2021-07-15

**Rating:** 6
**Confidence:** 5

**Summary:**

The submission deals with high resolution satellite images, given a low resolution image. The method is trained by learning to predict a high resolution image given a low resolution one, adversiarially, knowing the true high resolution. The point is to use LR imagery to predict HR at points in time where actual HR images are not available.

Experiments aim at quantifying the quality of generated images, both from a structural and reconstruction quality standpoint, as well as using the generated images in a potential downstream task of object counting, specifically, buildings and swimming pools.

**Limitations And Societal Impact:**

limitations are not very clearly discussed, while i do not see any potential adverse societal impact.

**Main Review:**

The paper deals with an interesting problem and surely deserves some attention. I think the method presented is robust and results encouraging, but I also have some concerns about practical scenarios and settings which maybe should be better presented and motivated in the manuscript.

In more detail:
- It is clear that the method aims at good reconstruction, but from the paper is somewhat unclear whether this can be extended from plain RGB to more wavelenghts. For many applications, having near infrared information is crucial, specifically crop monitoring and precision agriculture. In the paper, dataset only aim at reconstructing RGB images, which is a very fair target, but would it perform better if the LR is characterised also by near-infrared information, or if the target HR also contains NIR? I understand only using RGB makes more data available, but in the specific experiments, using sentinel data, other channels would be at hand.
- I would like to see a better distinction between learninng methods for superresolution. At the end of the day, one can train a SR model on LR-HR pairs, and use it to predict HR from a given LR. I see that standard SR methods do not make use of time, but from the plot in Fig. 5, it does not seem to _that_ relevant.
- Fig. 5. Is, in facts, statistically significant the difference of using or not using temporal information? I think that the model can struggle not only with ground cover changes (e.g. LL 210-212) but also with seasonal effects, atmospheric effects, sun-angle, etc. These aspects are not well considered in the manuscript and would be very beneficial to discuss these limitations.  The simples example would be the presence of occluding clouds or opaque clouds attenuating signals.
- In the same figure, I was expecting to see a spike in accuracy after roughly 1y, where the vegetation cycle is roughly at the same point. but I do see a spike at ~510 days, which is weird. What is the cause of that? only a noisy "by chance" spike, or something more? What about crop rotation for this dataset, could it influence the results? This spike is the only place where the time and not time models differ (but also larger error bars -- if the shaded area are std of experiment replica -- it is not explained).
- Another important point on the same line, is the argument about spatial resolution and the GSD. What is the maximum difference in GSD the model can handle? I am curious since for the swimming pools, LR image resolution (RGB Sentinel 2) often is larger than most small private swimming pools, where a given swimming pool is either contained in a pixel or diluted over a few. How can the model reconstruct such things, if those elements are not visible in the LR images? So I understand one should find best images for the downstream task, and it is obvious that one cannot reconstruct 10cm imagery from 250m GSD MODIS imagery, but some discussion around the issue would be welcome. I think that linking these to the amount of spectral information, could give hints about how to extend the work, since using infrared, for instance, could provide information about land covers not clearly distinguishable in RGB.
- It is mentioned that land cover changes affect results. This, coupled to the GSD problem of the point above, would suggest that there is a setting in which changes are modes of failures. Can this be a problem also for downstream tasks, and how can one prevent downstream task failures? For instance, not seeing swimming pools in LR data prevents them from being counted, which is a clear limitation.  Also, I can imagine that generating HR images far in the future, would greatly reduce confidence of the objects contained in the HR image. All in all, I wonder (since LR/HR pairs are rarely at uniform time intervals) how this can be detected at inference.

**Time Spent Reviewing:**

3

---

> ### Author Response · Authors · 2021-08-10
> **New results for w/ t v.s. w/o t comparison and clarifications**
>
> Thank you for your detailed review! We appreciate your constructive comments.
> 1. **NIR:** We certainly see the inclusion of additional bands as a compelling and natural extension of our work. Our method can be easily extended to include more bands by changing the number of input/output channels. We chose RGB bands because they are commonly available in remote sensing devices and they are sufficient for our targeted tasks. Thus we see our results as a lower bound of what can be achieved, and enhancements using additional bands are left to future work.
> 2. **Figure 5:**
>     - **Standard SR v.s. Our method:** We would like to first clarify that Figure 5 is NOT for the comparison between the standard SR model (trained on LR-HR pairs as suggested by the reviewer) and our model (trained on (LR, HR’)-HR tuples). The difference between w/ t and w/o t models is whether or not to learn the time embedding in the positional encoder E. Neither w/ t nor w/o t model is standard SR model and both are trained on (LR, HR’)-HR tuples. We will clarify this in the final version.
> Here we also train an additional standard SR version of our model using only LR-HR pairs on the Texas Housing dataset. As shown in the table below, our approach has clear advantages over the standard SR method. It is also worth noting that the LR images in our experiments are from real LR devices, which is different from synthetic LR images created by downsampling like many SR benchmarks. Leading standard SR methods such as DBPN and SRGAN do not perform well as shown in the experiments (Table 1 and 2).
> | EAD 	|  t’ > t 	|         	|        	|        	| t’ < t 	|         	|        	|        	|
> |---------	|:-------:	|:-------:	|:------:	|:------:	|:------:	|:-------:	|:------:	|:------:	|
> |         	|   SSIM  	|   PSNR  |  FSIM  	|  LPIPS 	|  SSIM  	|   PSNR  |  FSIM  	|  LPIPS 	|
> |   Ours  	| 0.6470  	| 22.4906 | 0.7904 	| 0.3695 	| 0.5225 	| 19.7675 | 0.7280 	| 0.4275 	|
> |  LR-HR  |  0.4731 	| 19.9537 | 0.7047 	| 0.4991 	| 0.3535 	| 17.4470 | 0.6453 	| 0.5400 	|
>     - **Statistical Significance:** w/ t is statistically significantly better than the w/o t model. Here we provide additional plots for SSIM, FSIM, and LPIPS results: https://ibb.co/ww46bx4. From the plots, we can see clear improvement in the w/t model.
> We use permutation tests with differences between means to test the statistical significance. FSIM and LPIPS achieve p-value 0.002 and 0 respectively for null hypothesis “w/ t is not better than w/o t”. PSNR achieves p-value=0.09 when |t’-t|>300 days, which shows that models with t have more benefits when working with longer time series as acknowledged and discussed in LL 244-246.
> Here we also report average results for both models and as we can observe, w/t model performs better in all metrics on fMoW-Sentinel2 Crop Field dataset. Hence learning t is crucial for our model performance and our conclusion in LL 242-246 does not change.
> |  EA64 	|  SSIM  	|   PSNR  	|  FSIM  	|  LPIPS 	|
> |:-----:	|:------:	|:-------:	|:------:	|:------:	|
> |  w/ t 	| 0.3905 	| 16.8879 	| 0.6827 	| 0.5197 	|
> | w/o t 	| 0.3844 	| 16.8075 	| 0.6774 	| 0.5380 	|
>     - **Spike around 510 Days:** The apparent spike might be due to fluctuations, as we test the null hypothesis “there is no spike around 510 days” and obtain p-value=0.21 for w/ t model and 0.22 for w/o t model with the same test described above.
> In addition, we do observe spikes at ~1y for all the other metrics: with null hypothesis “there is no spike around 1 year”, we obtain w/ t p-value=0.003, w/o t 0.005 for SSIM, w/ t 0.008, w/o t 0.01 for FSIM, and w/ t 0.002, w/o t 0.003 for LPIPS with the same tests described above. Notice that these metrics are not necessarily correlated, as discussed in [1].
> We do acknowledge the potential influence of crop rotation. However, since we do not have the crop type labels for the images, it is difficult to verify this effect.
>     - **Implementation Details:** We have included details and analysis about Figure 5 in Appendix F, which contains the explanation of the error bars. We will add reference to the appendix.
> 3. **Limitations and Failure Case Analysis:** We have discussed some limitations and failure cases in Appendix G. Furthermore, our model is still expected to suffer from bias if trained on data that is not representative of the target distribution. We leave quantification of such bias to future work.
> 4. **Small Objects in LR, LR GSD Limits and its Influence on Downstream Tasks:** It is not surprising that the model generates high resolution details because the model leverages a rich prior of what HR images look like acquired via the cGAN loss, and GANs are proven to be able to learn to generate high frequency details [2,3]. While using LR devices (downsampling) causes information loss, we still receive signals from LR pixels (e.g. swimming pools can still change pixel values in LR even though not necessarily humanly perceivable). Experiments in Figure 3 and Section 5.5 show that these signals are enough for our model to reliably reconstruct HR images that are high quality and applicable to downstream tasks.
> That being said, in more extreme scenarios (eg. LR has 250m GSD v.s. HR has 1m GSD), LR would provide very little information and therefore there would be much more uncertainty in the reconstruction. In this case, our model will still generate high resolution details, but they might not match the ground truth. We have begun experiments on the SpaceNet 7 dataset [4], an urban development dataset with dozens of locations across several continents with 4m GSD, paired with Landsat LR [5] with 30m GSD, and our method has worked for that as well. Here are some examples of samples from our EAD model on this dataset: https://ibb.co/gTLYkxn, https://ibb.co/ysGS6Y7.
> 5. **Dealing with Non-uniform Time Intervals and Confidence Score at Inference Time:** We would like to clarify that our model is applicable to time series of varying lengths with non-uniform time intervals (e.g. fMoW-Sentinel2 Crop Field dataset). In fact, this is one of the benefits of parametrizing models with a continuous time dimension. Our discussion about Figure 5 also shows that it is more beneficial to use our model than the baselines when working with longer time gaps.
> The key factor is the extent of changes between t and t’ rather than the time gap. In image pairs with greater changes, the information loss in LR manifests higher uncertainty, and thus the generator will produce more variegated details. We can potentially determine the confidence by calculating the degree of changes from t to t’, and the implementation is left for future work.
>
> **References:**
> [1] Richard Zhang, Phillip Isola, Alexei A. Efros, Eli Shechtman, Oliver Wang. The Unreasonable Effectiveness of Deep Features as a Perceptual Metric. CVPR, 2018
> [2] Tero Karras, Samuli Laine, Timo Aila. A Style-Based Generator Architecture for Generative Adversarial Networks. CVPR, 2019
> [3] Ivan Anokhin, Kirill Demochkin, Taras Khakhulin, Gleb Sterkin, Victor Lempitsky, Denis Korzhenkov. Image Generators with Conditionally-Independent Pixel Synthesis. CVPR, 2021
> [4] Adam Van Etten, Daniel Hogan, Jesus Martinez Manso, Jacob Shermeyer, Nicholas Weir, Ryan Lewis. The Multi-Temporal Urban Development SpaceNet Dataset. Proceedings of the IEEE/CVF Conference on Computer Vision and Pattern Recognition (CVPR), 2021, pp. 6398-6407
> [5] U.S. Geological Survey, 2016, Landsat—Earth observation satellites (ver. 1.2, April 2020): U.S. Geological Survey Fact Sheet 2015–3081, 4 p., https://doi.org/10.3133/fs20153081.

---

### Official Review · Reviewer_3V3Y · 2021-07-16

**Rating:** 5
**Confidence:** 5

**Summary:**

This paper presents a method to generate high-resolution satellite imagery from a pair of input low resolution image and a reference high resolution image of the same location but taken at a different time instant. The main problem the author(s) are addressing is the lack of high quality satellite imagery(specially temporally), and that can be resolved using multiple low resolution and a reference HR image. The authors claim that such a high resolution image will be useful for object counting tasks, specially over time when the intermediate high-res images are not available. Their model comprises of a feature mapper (for fusion of input low-res and reference high-res image),  feature encoder (for both the Fourier and spatial feature) and finally the synthesiser (inspired by nerf operating in the temporal domain keeping the spatial domain fixed) they curate a dataset from the publicly available datasets or services like google earth and perform their experiments on them. The authors also conduct a Human study for the verification of their approach along with the usual metrics for image similarity.

**Limitations And Societal Impact:**


Can the authors comment on hoe their system will cope with dense urban areas specially areas which have high population where objects in the contention (like buildings) are smaller than dataset. Also how well will this method generalise to different scene with differing urban landscape.

**Main Review:**

This paper proposes a interesting application of the already well researched problem if image super resolution. The super resolution is applied on the satellite images, where input to the system is a low resolution image and a reference high resolution image, but taken in a different time instant. The reference HR image provides as a guidance to for high quality realistic texture in the final synthesized image. Similar attempts have been done in general Single image SR ( e.g. SRNTT: Image Super-Resolution by Neural Texture Transfer, CVPR19).

In the paper the LR image corresponds to 10m/pixel (assuming in each spatial dimension) and the HR image is 1m/pixel. The method is trained using a single LR, reference and the target image. It is claimed that having multiple LR image as input does not add to the final reconstruction as the inputs low resolution images across time prove little additional information. However it seems counter-intuitive as you would expect the system to be able to bridge the gap better between the reference HR time t^{prime} and t. Have the authors tried it or there is lack of granular data to do the verification.

Can the authors elaborate more on the image registration between the input LR and input reference HR.  The reference HR image at 1m/pix resolution might contain objects like vehicles which might not be present in the LR time or not visible in the lower resolution. Does that have an impact on the overall performance of the system.

Can the authors provide some information on the ratio of the rectangles (bounded by the latitude/longitude, Appendix section C) of the are of interest  for the LR and the HR image. That give an indication of how much SR (e.g. 4x, 8x,..) is being done, and put this method in the perspective of the general task of Single image SR.

Can the authors please provide some additional experimental details like input image resolutions, output image resolutions (in spatial coordinates)

The paper is clear and easy to understand. Additional materials in Appendix are useful for understanding the experiments and the dataset.

Regarding the human evaluation of object counting task, the HR (at t^{prime}) is expected to be different at different time stamp (t). However for the building counting task, the reference HR(t^{prime}) image is significantly different than the target HR(t). This might make the generation task harder.  Can the author please comment on temporal granularity of reference and input images.

Can the authors please comment/discuss on the potential drawbacks of their systems. Some failure examples will be a good to demonstrate the limits of the system, and help further research. Further if they can please comment on how the proposed method will cope with large moving urban artifacts (often captured in satellite images ) like large vehicles, etc.

Can the authors please mention running performance (training times, inference times) for their method.

In fig. 4, there seems to be a slight colour distortion from the Ground Truth and methods EAD/EA64. However  the baseline methods are closer to ground truth in terms of colour temporature. Can the authors comment if this kind of distortion an atrifact of the system which is guided by reference HR image?

Their model has three part structure, first a feature mapper for combining the LR and ref HR images. Then their encoder combine Fourier the features (as proposed initially by [2]) to allow  their model to learn about temporal variation of the region. The authors take inspiration from the Neural radiance field to generate a temporally novel viewpoint while keeping the spatial dimension fixed. Thus the authors propose a system comprising of well known modules that work and have applied for satellite imagery, which is lacking in novelty. Finally, the resulting image is used for downstream object counting task, where it shows that human subjects perform better with the proposer sure-resolution method. However, this might not be the same for automated object counting.


**Time Spent Reviewing:**

4

---

> ### Author Response · Authors · 2021-08-10
> **New experiments, answers to specific questions and discussions**
>
> Thank you for your comprehensive review and constructive comments!
> 1. **SRNTT:** Thank you for suggesting SRNTT, we will add a discussion to our paper.
> While SRNTT also uses a HR reference image, it does not learn the additional time dimension to leverage the HR of the same object at a different time. In addition, our model uses a perceptron based generator while SRNTT uses a CNN based generator. Although we would love to compare with SRNTT, its authors only provide 4X SR, which is insufficient for our experiment, and upscaling it to larger SR ratios requires nontrivial modifications to the original networks.
> 2. **Multiple LR as Input:** Here we provide an experiment using our model with two additional LR image inputs from different time steps (Sentinel-2 images from 2017 and 2019) in Texas Housing dataset. We do not observe improvements from this setting. This agrees with our claim that given a LR image at the target time, other LR views from different time steps (from the past or future) provide little or no additional information.
> | EAD     	|  t’ > t 	|         	|        	|        	| t’ < t 	|         	|        	|        	|
> |-------------	|:-------:	|:-------:	|:------:	|:------:	|:------:	|:-------:	|:------:	|:------:	|
> |             	|   SSIM  	|   PSNR  	|  FSIM  	|  LPIPS 	|  SSIM  	|   PSNR  	|  FSIM  	|  LPIPS 	|
> |     Ours    	| 0.6470  	| 22.4906 	| 0.7904 	| 0.3695 	| 0.5225 	| 19.7675 	| 0.7280 	| 0.4275 	|
> | Multiple LR 	|  0.6266 	| 22.0939 	| 0.7814 	| 0.3873 	| 0.5023 	| 19.5255 	| 0.7198 	| 0.4428 	|
> 3. **Image Registration, SR scales and Dimensionality:** The SR scales can be calculated by the ratio of device resolutions (e.g. Sentinel-2 -> NAIP will be 10/1 or 10X) and the input/output resolutions are both 256X256 as mentioned in Appendix B LL 34. We extract LR and HR images using the same shape files/geo-coordinate bounding boxes. The resampling method mentioned in Section 4 LL 123-125 and Appendix C LL 70-73 is required for each pixel in the LR and HR to be geographically aligned. We will add references to the appendix  in the main paper.
> 4. **Small Objects in LR:** It is not surprising that the model generates high resolution details because the model leverages a rich prior of what HR images look like acquired via the cGAN loss, and GANs are proven to be able to learn to generate high frequency details [1,2]. While using LR devices (downsampling) causes information loss, we still receive signals from LR pixels (e.g. swimming pools can still change pixel values in LR even though not necessarily humanly perceivable). Experiments in Figure 3 and Section 5.5 show that these signals are enough for our model to reliably reconstruct HR images that are high quality and applicable to downstream tasks.
> That being said, in more extreme scenarios (eg. LR has 250m GSD v.s. HR has 1m GSD), LR would provide very little information and therefore there would be much more uncertainty in the reconstruction. In this case, our model will still generate high resolution details, but they might not match the ground truth. We have begun experimenting with datasets with different resolutions (details in **Applications to Urban Areas**) and we think it is an interesting future direction to explore.
> 5. **Temporal Granularity of Human Evaluation Experiment:** As discussed in LL 256-257, the object counting human evaluation task uses images randomly sampled from Texas Housing dataset with t’>t setting. As discussed in LL 164-165 and 204-205, our Texas Housing dataset consists of houses and their surrounding neighborhoods that are effectively built between 2014 and 2017. Therefore, we expect significant changes in real object count in many of these locations, and this task is designed to be challenging.
> 6. **Limitations:** Limitations and failure cases are discussed in Appendix G. In addition, our model is still expected to suffer from bias if trained on data that is not representative of the target distribution. We leave quantification of such bias to future work.
> 7. **Applications to Urban Areas:** Our model would work for dense urban areas as long as datasets with appropriate resolutions are available as analyzed in **Small Objects in LR**. We have started experiments on the SpaceNet 7 dataset [3] with GSD 4m, an urban development dataset with dozens of locations across several continents, paired with Landsat LR [4] with GSD 30m, and our method has worked for that as well. Here are some examples of samples from our EAD model on this dataset: https://ibb.co/gTLYkxn, https://ibb.co/ysGS6Y7.
> The challenge with generating moving artifacts stems from the dataset limitation rather than the model. For this task, we would require a training dataset that contains LR-HR pairs captured at exactly the same time. Since this is difficult to achieve in practice, we may leverage downsampled HR images in our training.
> 8. **Training/Inference Time:** We train each model to convergence which takes around 4-5 days on 1 NVIDIA Titan XP GPU. EAD can sample 1500 images in around 2.5 minutes (10 images/s) and EA64 can sample 1500 images in around 19 minutes (1.3 image/s). The difference in inference times results from the inference by patch technique described in Appendix A.1.
> 9. **Color Distortion:** We respectfully disagree with the comment on color distortion. Quantitative results including PSNR show that our model obtains higher pixel value accuracy across different datasets (Table 1 and 2), which agrees with the qualitative results in Figure 9 in the appendix and the third row in Figure 4. This implies that our model has less color distortion than the baselines.
> 10. **Novelty:** We believe our method is novel. Firstly, to the best of our knowledge, no other NeRF-inspired models have been applied to satellite imagery time series before.
> Secondly, as discussed at line 77-86, our model is a conditional generative model for paired image time series, which is different from all relevant NeRF-inspired models we have found in literature [2,8,9,10,11,12]. Our model architecture is novel in that: it is a pixel synthesis model that learns a conditional 2D spatial coordinate grid along with a continuous time dimension, which is tailored for remote sensing where the same location/object can be captured by different devices (eg. NAIP or Sentinel-2) at different times (eg. year 2016 or 2018).
> It is worth noting that our LR images are captured by remote sensing devices (e.g. Sentinel-2), which is different from synthetic LR images created by downsampling like many SR benchmarks. As we have shown in our experiments, leading SR models such as DBPN and SRGAN do not perform well in this setting.
> 11. **Automated v.s. Human Counting:** We believe that human evaluation reflects what in principle can be achieved (gold standard) with models learned from annotations. As mentioned in LL 267-268, human level object counting in satellite images is still largely an open problem. As such, we believe that human evaluation yields a meaningful measurement of our model performance. Furthermore, there are many applications involving manual analysis of the satellite images (e.g. crowdsourcing earthquake damage assessment [7]).
>
> **References:**
> [1] Tero Karras, Samuli Laine, Timo Aila. A Style-Based Generator Architecture for Generative Adversarial Networks. CVPR, 2019
> [2] Ivan Anokhin, Kirill Demochkin, Taras Khakhulin, Gleb Sterkin, Victor Lempitsky, Denis Korzhenkov. Image Generators with Conditionally-Independent Pixel Synthesis. CVPR, 2021
> [3] Adam Van Etten, Daniel Hogan, Jesus Martinez Manso, Jacob Shermeyer, Nicholas Weir, Ryan Lewis. The Multi-Temporal Urban Development SpaceNet Dataset. Proceedings of the IEEE/CVF Conference on Computer Vision and Pattern Recognition (CVPR), 2021, pp. 6398-6407
> [4] U.S. Geological Survey, 2016, Landsat—Earth observation satellites (ver. 1.2, April 2020): U.S. Geological Survey Fact Sheet 2015–3081, 4 p., https://doi.org/10.3133/fs20153081.
> [5] Tero Karras, Samuli Laine, Miika Aittala, Janne Hellsten, Jaakko Lehtinen, and Timo Aila. Analyzing and improving the image quality of StyleGAN. InProc. CVPR, 2020.
> [6] Dmitry Ulyanov, Andrea Vedaldi, and Victor Lempitsky. Deep image prior. arXiv:1711.10925,4002017.
> [7] Barrington L, Ghosh S, Greene M, Har-Noy S, Berger J, Gill S, Lin AY-M, Huyck C. Crowdsourcing earthquake damage assessment using remote sensing imagery. Ann. Geophys.
> [8] Zhengqi Li, Simon Niklaus, Noah Snavely, and Oliver Wang. Neural scene flow fields for space-time view synthesis of dynamic scenes. https://arxiv.org/abs/2011.13084, 2020.
> [9] Ben Mildenhall, Pratul P. Srinivasan, Matthew Tancik, Jonathan T. Barron, Ravi Ramamoorthi, and Ren Ng. Nerf: Representing scenes as neural radiance fields for view synthesis. In ECCV, 2020
> [10] Katja Schwarz, Yiyi Liao, Michael Niemeyer, and Andreas Geiger. Graf: Generative radiance fields for 3d-aware image synthesis. In Advances in Neural Information Processing Systems (NeurIPS), 2020.
> [11] Wenqi Xian, Jia-Bin Huang, Johannes Kopf, and Changil Kim. Space-time neural irradiance fields for free-viewpoint video. https://arxiv.org/abs/2011.12950, 2020.
> [12] Alex Yu, Vickie Ye, Matthew Tancik, and Angjoo Kanazawa. pixelNeRF: Neural radiance fields from one or few images. In CVPR, 2021.

---

### Official Review · Reviewer_AEn7 · 2021-07-16

**Rating:** 8
**Confidence:** 4

**Summary:**

This paper proposes using a NeRF-like approach for generating high-resolution imagery at some time t based off of high-resolution imagery at time t' and low-resolution imagery at t. This is a practically useful application for different problems in conservation and sustainability as the authors note. They compare the proposed method with SR based methods and image fusion models (e.g. cGAN and Pix2Pix) on two datasets.

The approach consists of:
1.) An image encoder that takes the conactenated HR and LR images as input
2.) A decoder that produces pixel level features
3.) A learned positional embedding
4.) A pixel-wise decoder that produces the desired high-resolution imagery from the pixel level features and position embedding
5.) A discriminator network

The entire process is trained using a cGAN loss and L1 reconstruction loss.

Evaluation is performed with SSIM, FSIM, and LPIPs.

**Limitations And Societal Impact:**

The authors address the limitations and potential negative societal impacts. While they acknowledge that their method has potential applications in surveillance, high-resolution on demand satellite imagery is both commercially available and available to government level entities, therefore the additional negative impacts of their method are minimal.

**Main Review:**

This is a well written paper with very clear experiments and contributions with immediate practical application in remote sensing.

Questions for the authors:
- Is LPIPs with VGG features acceptable in remote sensing contexts? What is the reason that VGG features from ImageNet should be able to measure visual perception of high-resolution imagery?
- Did you consider using the NIR bands from Sentinel / NAIP? These spectral values may better constrain the output of the model
- What is the motivation for EA64 vs. EAD?
- Is it possible to evaluate counting based downstream tasks on fMoW?

## Minor Comments

- Line 126-127, should one of the three F_E be F_A and another be F_D (matching Figure 2)?

**Time Spent Reviewing:**

2

---

> ### Author Response · Authors · 2021-08-10
> **Thank you for your review**
>
> Thank you for your detailed review and suggestions! We appreciate your positive comments and provide the following answers to your questions.
> 1. **LPIPS:** LPIPS with VGG features has been used in remote sensing contexts in other work [1], and this metric is proven to be suitable for assessing superresolution (SR) reconstruction image quality and is claimed to be closer to human perception [2]. We use this in conjunction with other metrics (SSIM, PSNR, FSIM) to provide a more comprehensive evaluation of image quality.
> 2. **NIR:** We certainly see the inclusion of additional bands as a compelling and natural extension of our work. Our method can be easily extended to include more bands by changing the number of input channels. We chose RGB bands because they are commonly available in remote sensing devices and they are sufficient for our targeted tasks. Thus we see our results as a lower bound of what can be achieved, and enhancements using additional bands are left to future work.
> 3. **EAD v.s. EA64:** The motivation for EA64 v.s. EAD is to examine the capabilities of models with and without spatial downsampling/upsampling. As we mentioned in Section 5.2 LL 181-183 and Appendix A LL 10-15, EA64 can be considered as a model without spatial downsampling/upsampling since F_E has stride = 1 and F_D is the identity function. As we discussed in LL 236-241, each model has advantages and disadvantages.
> 4. **Counting Tasks on fMoW:** fMoW is designed to include one main object in one image. Although counting based tasks may technically still be possible on certain categories, fMoW does not have ground truth object count labels so it is difficult for us to quantitatively evaluate our models on this specific task.
> 5. **Minor Typos:** Thank you for pointing out the typos! We will fix the errors in our paper.
>
> **References:**
> [1] Gong, Y.; Liao, P.; Zhang, X.; Zhang, L.; Chen, G.; Zhu, K.; Tan, X.; Lv, Z. Enlighten-GAN for Super Resolution Reconstruction in Mid-Resolution Remote Sensing Images. Remote Sens. 2021, 13, 1104. https://doi.org/10.3390/rs13061104.
> [2] Zhang, R.; Isola, P.; Efros, A.A.; Shechtman, E.; Wang, O. The unreasonable effectiveness of deep features as a perceptual metric. In Proceedings of the IEEE Conference on Computer Vision and Pattern Recognition, Salt Lake City, UT, USA, 18–22 June 2018; pp. 586–595.

---

### Decision · Program_Chairs · 2021-09-27

**Decision:**

Accept (Poster)

**Comment:**

Three of the four reviewers recommended accepting the paper , and one of them increased the score following the rebuttal. I am happy to accept it, I encourage the authors to include the additional material that they discussed in the rebuttal in the final version.